# Structure-Based Long-Term Biodegradation of the Azo Dye: Insights from the Bacterial Community Succession and Efficiency Comparison

Chao Zhu [1,*], Zarak Mahmood [2], Muhammad Saboor Siddique [3], Heyou Wang [1], He Anqi [1] and Mika Sillanpää [4,5]

1   School of Environmental Science & Engineering, Shaanxi University of Science and Technology, Xi'an 710021, China; 1903054@sust.edu.cn (H.W.); 1903020@sust.edu.cn (H.A.)
2   School of Ecology and Environment, Northwestern Polytechnical University, Chang'an District, Xi'an 710129, China; dr.zarak@nwpu.edu.cn
3   Key Laboratory of Drinking Water Science and Technology, Research Center for Eco-Environmental Sciences, Chinese Academy of Sciences, Beijing 100085, China; saboorsiddique.uaf@gmail.com
4   Faculty of Environment and Labour Safety, Ton Duc Thang University, Ho Chi Minh City 700000, Vietnam; mikaesillanpaa@gmail.com
5   Environmental Engineering and Management Research Group, Ton Duc Thang University, Ho Chi Minh City 700000, Vietnam
*   Correspondence: zhuchao@sust.edu.cn

**Abstract:** In this study, microbial community dynamics were explored during biological degradation of azo dyes with different chemical structures. The effect of the different molecular structures of the azo dyes was also assessed against the simultaneous removal of color and the bacterial community. Winogradsky columns were inoculated with dewatered sludge and separately fed with six different azo dyes to conduct the sludge acclimatization process, and nine bacterial decolorizing strains were isolated and identified. The decolorization and biodegradation performances of the acclimated system and isolated strains were also determined. Results showed that the bacterial isolates involved in decolorization and the degradation of the azo dyes were mainly associated with the azo dye structure. After 24 h acclimatization at room temperature without specific illumination, immediate decolorization of methyl red (89%) and methyl orange (78%) was observed, due to their simple structure compared to tartrazine (73%). However, after 8 days of acclimatization, methyl red was easily decolorized up to 99%, and about 87% decolorization was observed for orange G (87%), due to its complex chemical structure. Higher degrees of degradation and decolorization were achieved with *Pseudomonas geniculate* strain Ka38 (Proteobacteria), *Bacillus cereus* strain 1FFF (Firmicutes) and *Klebsiella variicola* strain RVEV3 (Proteobacteria) with continuous shaking at 30 °C. The azo dyes with benzene rings were found to be easier to decolorize and degrade with similar microbial communities. Moreover, it seems that the chemical structures of the azo dyes, in a sense, drove the divergent succession of the bacterial community while reducing the diversity. This study gives a deep insight into the feasible structure-based artificial manipulation of bacterial communities and offers theoretical guidance for decolorizing azo dyes with mixed bacteria cultures.

**Keywords:** azo dye; decolorization; biodegradation; gene metabarcoding; bacterial community

## 1. Introduction

Azo dyes containing one or more azo bonds(–N=N–), represent the largest class of dyes used in the textile industry, because of their low cost, easy synthesis, high stability and wide variety of available colors compared to natural dyes [1]. According to an estimate, 10 million tons of these dyes are consumed annually, costing about USD 31 billion [2]. Azo dyes and their by-products are aromatic in nature, having an azo group, and they are toxic to aquatic organisms [3]. About 10−15% of azo dyes are discharged directly

into the aqueous environment without any treatment during the manufacturing and dying processes [4], despite the fact that they have been found to be toxic to the aquatic environment [5]. Even minor amounts of azo dyes present in water bodies can significantly affect photosynthetic activity by diminishing the light penetration and can affect aesthetic aspects and the quality of the water [6]. Therefore, in the recent past, the treatment of effluents containing synthetic dyes has gained much attention among scientists [7].

In contrast to the conventional physico-chemical treatment methods for wastewater containing azo dyes, previous studies have also reported that biological treatment approaches have gained in popularity due to their low cost, lower sludge production and more environmentally friendly behavior [8]. For instance, Mezohegyi et al. [8] studied the anaerobic decolorization of the azo dye Acid Orange in a continuous up-flow stirred packed-bed reactor (USPBR) filled with biological activated carbon (BAC) and reported about 96% bioconversion of the azo dye in 0.5 min. Similarly, Balapure et al. [9] reported complete decolorization and degradation of RB160 (100 mg/L) within 4 h, along with co-metabolism of yeast extract (0.5%), by enriched mixed cultures BDN (BDN). The entry of azo dyes into the ecosystem also has a significant impact on microbial communities. During the decolorization of azo dyes, the composition, activity and stability of the microbial community play an important part. Yu et al. [10] evaluated the relationship between the decolorization capacity of methyl orange and the microbial community structure and claimed about 85% and 75% of COD and methyl orange removal during the whole operational period, respectively. According to the literature, it is believed that the microbial community structure and diversity may significantly affect the performance and stability of biological processes (e.g., dye degradation) as reported elsewhere [11]. Qualitative and quantitative analysis of bacterial communities can also provide information for optimizing treatment processes [12,13]. In addition, azo dyes with various chemical structures and molecular weights were observed to exhibit different removal and decolorization rates [14]. For example, dyes with a simple structure and a lower molecular weight exhibit a higher decolorization rate and a lower removal rate. Conversely, the aforementioned complex structures of azo dyes result in different decolorization abilities of microorganisms.

Conventionally, the mechanism of microbial decolorization of azo dyes includes adsorption decolorization, enzymatic degradation and a combination of these two technologies [15,16]. Recently, the microorganisms used to absorb decolorized azo dyes are algae, yeasts, filamentous fungi and bacteria [17]. Generally, the adsorption capabilities of microorganisms mainly depend on proteins, lipids, sugars and other macromolecular substances, as well as the various functional groups in molecules (e.g., amino, carboxy, hydroxy and phosphate groups). These functional groups adsorb the azo dyes based on polarity, hydrogen bonds or electrostatic bonding. Proper pretreatment of microorganisms optimizes their adsorption properties. For example, addition of acids, formaldehyde, NaOH, $NaHCO_3$ or $CaCl_2$ to sterilizing microorganisms enhances the binding sites for adsorption. Currently, researchers are mainly focusing on microbial community diversity and community structures for the decolorization of textile effluent. Additionally, the evaluation of the microbial community structure against different by-products of the dyes can also help to identify the critical microorganisms and mechanisms associated with the transformation process. To best of our knowledge, in-depth evaluations of functional microorganisms and microbial communities against the degradation mechanisms of different azo dyes have still not been performed. Hence, issues regarding the response of the microbial community when applied to the degradation of azo dyes should be explored on the basis of microbial community structure, to further enhance performance efficiency.

Since dewatered sludge has a lower environmental water activity than normal activated sludge and often contains heavy metals [18], the microorganisms contained in it can often resist harsh environments such as high temperature, high salinity and a high concentration of heavy metals. Therefore, the use of dewatered sludge as a separation source to screen functional flora for their ability to decolorize azo dyes is of great value. At present, the effect of dehydrated sludge on the decolorization of azo dyes is still unknown.

Therefore, this study employed dewatered sludge as a new source of bacterial colony separation for decolorizing azo dyes, in order to screen functional bacteria for decolorizing azo dyes.

In this study, we explored the effect of the molecular structures on the biodegradation rate of six different azo dyes, using nine bacterial strains isolated from the dewatered activated sludge. In addition, the microbial habitat was investigated using isolation culture approaches and terminal restriction fragment length polymorphism (T-RFLP) after contact with various azo dyes. Lastly, a canonical correspondence analysis (CCA) was utilized to analyze the relationship between the microbial habitat and various factors in the different domestication treatments.

## 2. Materials and Methods

### 2.1. Activated Sludge

The sample of activated sludge was collected from the surplus sludge tanks of the full-scale municipal wastewater treatment plant (MWTP) located in Xi'an (34°38′20.89″ N 109°1′14.47″ E) [19]. A sequential batch-type intermittent feeding operation mode was adopted for sludge dewatering to further treat the various azo dyes [18].

### 2.2. Dyes and Reagents

Methyl orange, methyl red, orange I, orange G, tartrazine and alizarin yellow R sodium salt (Table S1) were selected as the model azo dyes because of their common use in the textile industry; they were purchased from Sinopharm Chemical Reagent Co., Ltd. (Shanghai, China). The biochemical reagents were procured from Beijing CW Biotech Co., Ltd. (Beijing, China). All other reagents used in this study were of analytical grade.

### 2.3. Sludge Acclimatization

Acclimatization experiments were performed in Winogradsky columns with a height, diameter and thickness of 99.0 cm, 5 cm and 1 cm, respectively (Figure 1). The dewatered sludge (100 g) was inoculated into acclimatized medium (800 mL) containing 30 mg/L of single azo dye in each of six different columns and one column with a mixture of the six azo dyes mentioned above, at room temperature without specific illumination. The selective culture medium contained $Na_2SO_4$ (5.0 g/L), $CaCO_3$ (2.5 g/L), microcrystalline cellulose (7.0 g/L), peptone (10.0 g/L) and NaCl (5.0 g/L) at neutral pH. About 10.0 g/L of yeast extract was used as a nutrient. After complete decolorization, the supernatant was removed and a new culture medium and dye added for the next acclimatization cycle. In total, 3 cycles were performed. At predetermined time intervals (about 7 days per cycle), 2 mL samples were taken, filtered and centrifuged, and the dye concentration was spectrophotometrically determined in UV2300 spectrophotometer (Tianmei, Ltd., Shanghai, China). Partial sludge samples of raw dewatered sludge and samples after acclimatization (21 days) were taken and kept at −20 °C for T-RFLP analysis.

### 2.4. Strain Isolation

About 10 mL of domesticated sludge from the Winogradsky column was transferred into 90 mL of sterile water (containing glass beads) followed by rapid mixing (10 min), to obtain a solution with a dilution of $10^{-1}$. Then, 1 mL of bacterial liquid (concentration $10^{-1}$) was placed in 9 mL of sterile water and mixed well to obtain a concentration of bacterial liquid of $10^{-2}$. The same method was adopted to prepare dilutions up to $10^{-8}$. Then, 0.5 mL of bacterial solution sampled from the $10^{-6}$, $10^{-7}$ and $10^{-8}$ dilution test tubes was applied to a solid separation medium and incubated for 24 h at 35 °C. A photograph of some of the colonies is shown in Figure 2, along with the SEM image. Different forms of single colonies were selected to inoculate into the liquid separation medium at 30 °C for 12 h, to extend the culturing and purification. Finally, the cultured bacteria liquid was placed on a solid separation medium to obtain a pure species single colony, for the dye culture medium decolorization experiments.

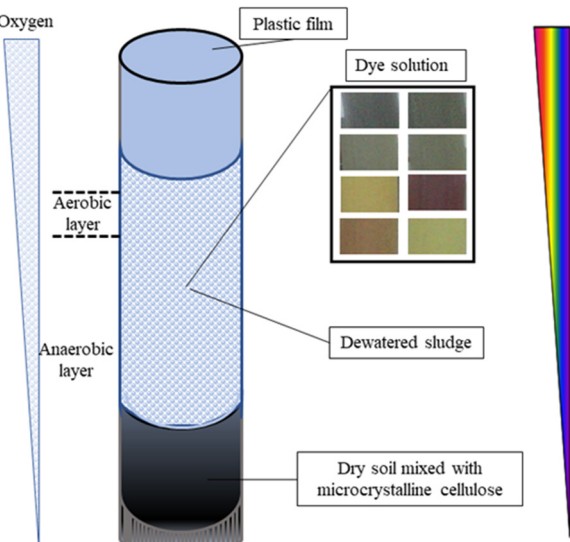

**Figure 1.** Experimental setup: Winogradsky column showing aerobic and anaerobic layers of dewatered sludge zone. Eight different dye solutions (methyl orange, methyl red, orange I, orange G, tartrazine, alizarin yellow R sodium salt, mixture and control) were used to inoculate the dewatered sludge. The rainbow spectrum shows the light penetration.

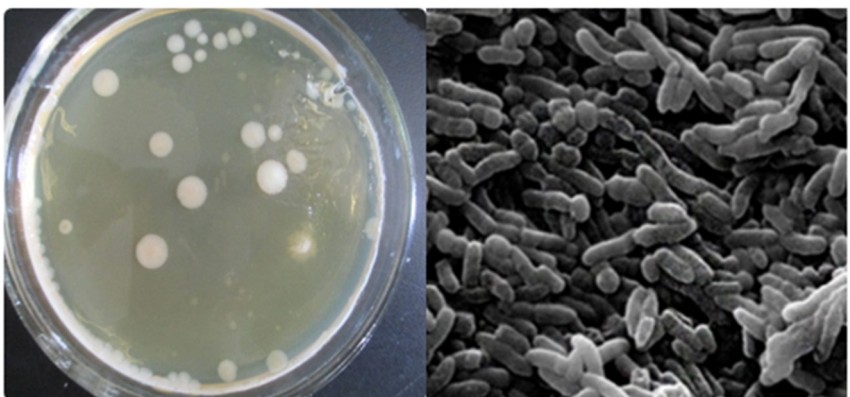

**Figure 2.** Partial isolated colonies from activated sludge after azo dye acclimatization. The right-hand photo presents the SEM image of one of the isolated strains.

In total, 9 strains were finally isolated and coded as T-1, T-2, T-3, T-4, T-5, T-6, T-7, T-8 and T-9. Their morphological and physiological–biochemical characteristics are presented in the supplementary information (Tables S2 and S3).

*2.5. Strain Identification*

The morphological, physiological and biochemical characteristics of the strains extracted from the acclimated sludge were determined with reference to the method previously reported in [20]. Genomic DNA was extracted from the acclimatized sludge using a SoilGen DNA Kit, following the manufacturer's protocol. The purified DNA samples of the strains were used as templates for polymerase chain reaction (PCR) amplification of the 16S rRNA gene using the bacterial community. The PCR amplifications were performed using 27F and 1492R as the forward and reverse primers, respectively. The PCR mixtures (50 µL) contained 5 µL of 10 × PCR buffer, 3.2 µL of dNTP (2.5 mM), 0.4 µL of rTaq (5 U/µL) 2 µL of Fam-27F (5 mM) and 50 ng of genomic DNA, added to 50 µL of ddH$_2$O. The following PCR program was used on a TGradient PCR cycler (Biometra Germany): 10 min at 94 °C, 30 s at 94 °C, 30 s at 55 °C, 45 s at 72 °C, 30 cycles and a final extension of 10 min at 72 °C. The PCR products were then visualized on 1.8% agarose gel stained with ethidium bromide. After

detecting the size of the amplified product by agarose gel electrophoresis, TA (thymine and adenine) cloning and sequencing was performed. Subsequently, the results of the sequenced 16S rRNA genes were compared with the related 16S rRNA gene sequence in GenBank. For similarity comparison analysis, Clustal X and MEGA5.1 software was used to construct the phylogenetic tree (Figure S1).

### 2.6. Decolorization Test of Isolated Strains

The isolated colonies were inoculated into 30 mL of liquid medium under aseptic operating conditions and cultured for 12 h with continuous shaking until the $OD_{600}$ was about 0.6. After culturing, 3 mL of cultured solution of each colony was separately inoculated into 150 mL of dye medium with continuous mixing at 30 °C. Subsequently, 2 mL of inoculated suspension was collected and centrifuged at 10,000 r/min for 5 min. The collected samples (1 mL) were then utilized to determine the UV–Vis absorbance at the maximum absorption wavelength of the azo dye. UV–Vis wavelength scanning was registered within the range of 200–800 nm. The percentage (%) of decolorization was calculated using Equation (1):

$$\text{Decolorization } (\%) = \frac{A(t_0) - A(t)}{A(t_0)} \times 100 \qquad (1)$$

where $A(t_0)$ and $A(t)$ are the initial absorbance intensity (i.e., time = 0 h) and the absorbance intensity after a particular reaction time (i.e., time = t), respectively. However, the effect of mixed inoculated dyes in one solution was also separately observed. All isolated strains were cultured in 30 mL of liquid medium at 30 °C with continuous shaking for 12 h. The mixed dyes solution contained methyl orange, golden orange I, orange yellow G, hydrazine yellow, and alizarin R sodium salt, with a concentration of 30 mg/L for each dye.

### 2.7. T-RFLP Analysis of Acclimated Sludge

A terminal restriction fragment length polymorphism (T-RFLP) analysis was used to reveal and compare the bacterial communities of all acclimatized sludge and raw dewatered sludge samples [18]. The genetic profiles of the amplified bacterial 16S rRNA were generated by restricted enzyme digestion. A soil DNA kit (Beijing Kangwei Century Biotech Co. Ltd., Beijing, China) was utilized for the DNA extraction from acclimatized activated sludge with different azo dyes. The reaction mixture (25 μL) contained 1 μL of each purified product as a template, 12.5 μL of 2 × Taq Plus PCR master mix, 9.5 μm of ddH$_2$O and 1 μm of reaction buffer (27F/1492R). The reaction was carried out at 95 °C—5 min, 94 °C—1 min, 60 °C—45 s and 72 °C—1 min for 25 cycles. Size separation of terminal restriction fragments was performed by capillary gel electrophoresis using an ABI gene analyzer 3130XL (Hitachi/Applied Biosystems, Foster City, CA, USA) equipped with capillaries loaded with POP$_4$ polymer (Thermo Fisher Scientific, Waltham, MA, USA). The relative migration of restricted fragments was determined via Genemarker HIV V1.7 (SoftGenetics Inc., State College, PA, USA). Finally, by removing primer peaks (>50 bps) and spurious peaks (relative abundance > 0.5%), the T-RFLP profile was determined. The relative peak area of a single T-RF was calculated according to the following equation:

$$\text{Ap} = Ni/N \times 100 \qquad (2)$$

where Ni and N are the peak areas of the single and total T-RFs, respectively. T-RFs were analyzed via MICA3 PAT program (available at http://mica.ibest.uidaho.edu/pat.php accessed on 8 August 2021), from which bacterial community structure spectra were constructed.

### 2.8. CCA Analysis

A correlation analysis was conducted on the microbial diversity fluctuations and taxonomical structure data of the microbial communities in response to the biodegradation of dyes, using canonical correspondence analysis (CCA) with Canoco (Windows 4.5 package).

In addition, the statistical analysis correlated the diversity of community-based species with environmental variations. Therefore, this study represents an extended survey of the microbial community response, distribution and abundance due to environmental variation. The analysis performed was based on the pH and COD values, percentage decolorization, evenness and Shannon diversity index, using canonical correspondence analysis (CCA). Parameters such as COD, pH, decolorization rate, evenness and Shannon diversity index were selected as variable factors for the seven different samples taken from the reactor. The CCA biplot characterizes the biological communities relative to the selected variables. The arrows of the biplot represent the variable factors, with the length of the arrow representing the maximal value at its tip [21].

## 3. Results and Discussion

### 3.1. Decolorization of Azo Dyes

The effect of the time interval on azo dye decolorization was tested for a range of 1 to 8 days, and the data were categorized into three groups (first, second and third group) according to the time interval. The first group samples were taken after 24 h, and the samples for the second group were taken after eight days. The acclimatization of the second group was performed to analyze the absorbance after eight days. After 24 h acclimatization, high decolorization efficiencies were observed for methyl red and orange G (Figure 3), with average color removal values of 90% and 88.5%, respectively. However, the performance was lower in the case of tartrazine. In addition, after 8 days of acclimatization, a high removal efficiency for methyl orange (98.5%) was observed, as a simultaneous effect, in comparison with all other dyes. The higher decolorization percentage at this stage was mainly ascribed to the higher growth rate of microorganisms in activated sludge [22]. Conversely, at this stage, the lowest performance was observed for orange G compared to all the other targeted dyes. The acclimatization period of the third group was same as that of the second group, but it was performed to analyze the UV–Vis absorbance. The average decolorization percentages of the mixture were 76%, 88% and 95% for the first, second and third stages (according to the time interval). The decolorization percentage for the third acclimatization stage was higher than for the first and second stages [23].

In comparison with the dye mixture, uniform and immediate decolorization was observed for all azo dyes with a simple structure and a lower molecular weight. However, a low percentage decolorization was observed for the azo dyes containing electron-withdrawing groups such as $-SO_3H$ and $-SO_2NH_2$ [14]. The structure of methyl orange was found to be simpler than those of orange I and orange G, because the N atom of orange I contains a naphthalene ring [24] and the orange G molecule contains an $SO_3^-$ group [25]. The presence of a naphthalene ring or $SO_3^-$ group also enhanced the absorption spectrum at high energy [26]. A similar study on the decolorization effect on dyes was conducted by the authors of [17], pointing out the increased detrimental effect on the degradation rate for the azo dyes containing electron-withdrawing groups such as -SO3-, compared to the azo dyes containing electron-donating groups such as $-NH_2$. The presence of electron-withdrawing groups on the benzene ring of the azo dye not only reduces the electron density but also weakens its -N=N- conjugated system, resulting in reduced active sites for the electrons.

The concentration effect of methyl orange on the percentage decolorization was analyzed within the range of 100 mg/L to 400 mg/L (Figure S2). Lower decolorization was observed at higher concentrations, i.e., 90%, 83% and 80% with 200 mg/L, 300 mg/L and 400 mg/L, respectively. In addition, the higher concentrations also covered the active sites of the azo reductase, causing a reduced decolorization and degradation rate. According to these findings, azo dyes with a concentration of 100 mg/L were utilized for further analysis.

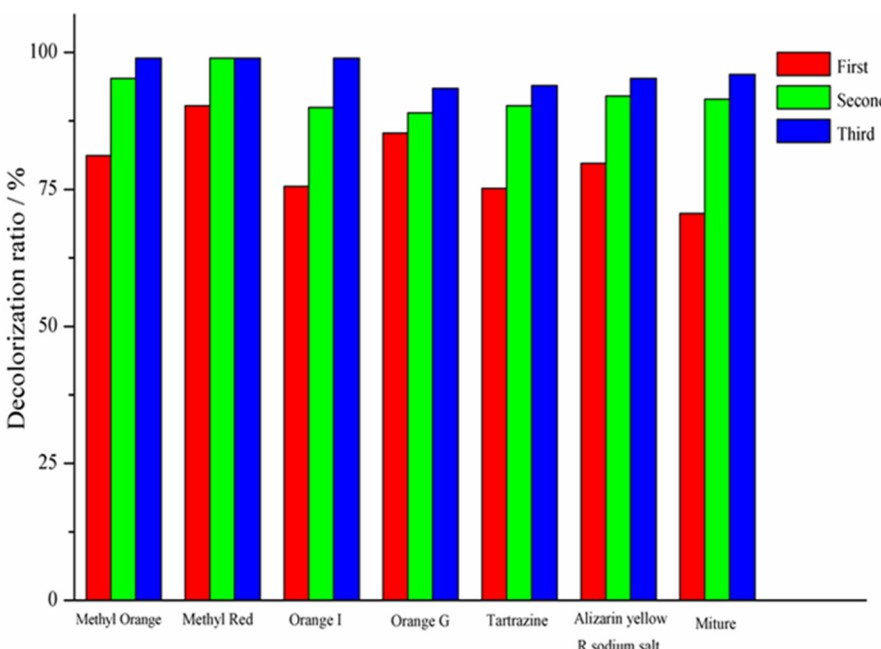

**Figure 3.** The percentage (%) decolorization of different azo dyes in activated sludge acclimatization systems for three groups. The first group had an acclimatization period of 24 h and the second and third groups had acclimatization periods of 8 days. For better understanding, a color view is recommended.

### 3.2. Cultured Bacterial Composition of Azo Dye in Acclimatized Sludge

After acclimatization in the Winogradsky columns, a total of nine typical bacterial strains were isolated from seven sludge samples with different azo dyes, reflecting a clear difference in the bacterial community structure (Figure 4). The bacterial isolates were identified by 16S rDNA sequencing. *Pseudomonas geniculate* strain Ka38 (Proteobacteria) were found to be the dominant bacteria for the decolorization process of methyl orange, accounting for 50% of the isolated and cultured strains. *Bacillus cereus* strain 1FFF (Firmicutes) had proportions of 60%, 38% and 35%, followed by *Exiguobacterium aestuarii* strain YS-6 (Firmicutes) and *Klebsiella variicola* strain RVEV3 (Proteobacteria) in orange I, alizarin yellow R sodium salt and the mixture, respectively. The dominant bacteria for orange G, tartrazine and the mixture of dyes were *Klebsiella variicola* strain RVEV3 (Proteobacteria) with proportions of 31.82%, 37.04% and 34.63%, respectively, followed by *Exiguobacterium aestuarii* strain YS-6 (Firmicutes) and *Bacillus cereus* strain 1FFF (Firmicutes). Sequence alignment using the NCBI database further showed that the isolates obtained in this study were mainly close relatives of *Pseudomonas* sp., *Stenotrophomonas* sp., *Bacillus* sp. and *Paracoccus* sp., which were distributed within *Firmicutes* and *Proteobacteria*. The adaptive changes that microorganisms experienced under domestication conditions included phenotypic adaptation (no change in the gene type of the population, which relied only on cell morphology or physiological changes to adapt to domestication conditions) and evolutionary adaptation (under the domestication pressure, the genetic type of the microorganism changed and new gene types appeared) [27].

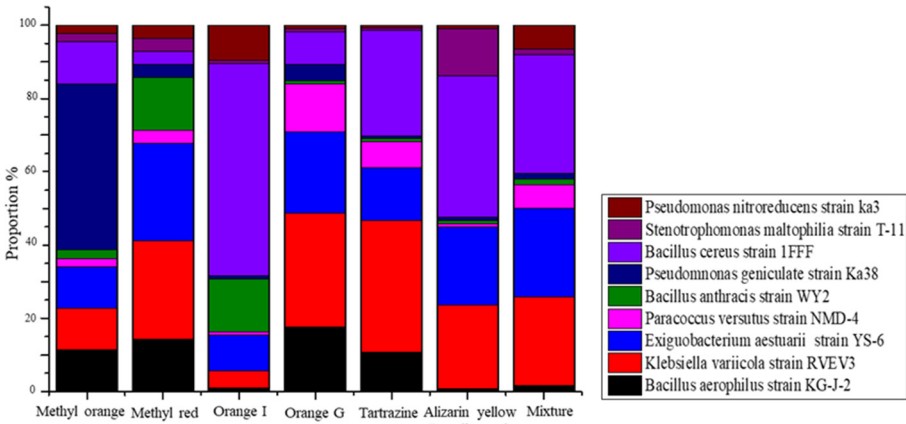

**Figure 4.** Comparison of cultured microbial community structures from activated sludge after acclimatization with different azo dyes. For better understanding, a color view is recommended.

The strain T-1 was homologous to *Bacillus anthracis* KG-J-2, having a phylogenetic similarity of 97%. Strain T-2 was highly homologous to *Klebsiella variicola* strain RVEV3, with a sequence similarity of 95%. The phylogenetic similarity between T-3 and *Exiguobacterium aestuarii* YS-6, and T-4 and *Paracoccus versutus* NMD-4 was 90% and 99%, respectively. The phylogenetic similarity between T-5 and *Bacillus anthracis* strain WY2 was only 83%. Strain T-6 was highly homologous to *Pseudomonas geniculate* ka38 with a phylogenetic similarity of 96%. Additionally, about 97% and 99% phylogenic similarity was observed between T-7 and *Bacillus cereus* strain BVC79, and T-8 and *Stenotrophomonas maltophilia* T-11, respectively. Similarly, T-9 showed a 99% phylogenetic similarity to *Pseudomonas nitroreducens* H-3 and *Stenotrophomonas chelatiphaga* ka32. These results proved that after proper domestication, a more functionally mutated population can be obtained from the indigenous flora of dewatered sludge. The presence of the above-mentioned microbial populations can play a promising role in dye decolorization, for dyes that can be found in the effluent of any textile industry. A total of seven isolates out of the nine obtained in this study showed a similarity percentage of less than 99% with the standard strains. Consequently, these strains can possibly be claimed as new species, with applicability in taxonomic studies. Bacterial isolates having a similarity less than 95% can probably be considered as novel genera. The polygenetic study placed these types of strains in a different branch, next to the closest relatives given by the GenBank database. This clearly indicated the survival of diverse group of microorganisms in dye-contaminated sludge, along with the utilization of the dye and xenobiotic compounds as energy sources.

The genus-level distribution was found to be different for each azo dye and mixture. The dominant populations were *Bacillus cereus* strain 1FFF, *Klebsiella variicola* strain RVEV3 and *Exiguobacterium aestuarii* strain YS-6. The most dominant group in the orange I, alizarin yellow R sodium salt and mixed dyes was *Bacillus cereus* strain 1FFF, with a compositional proportion of 60%, 38% and 35%, respectively. Similarly, the *Klebsiella variicola* strain RVEV3 was found to be dominant in methyl red, orange G, tartrazine, alizarin yellow R sodium salt and the mixed dyes, with a compositional proportion of 30%, 25%, 40%, 22% and 23%, respectively. The *Exiguobacterium aestuarii* strain YS-6 was observed in methyl orange (30%), alizarin yellow R sodium salt (25%) and the mixture of dyes (30%). *Pseudomonas nitroreducens* strain Ka3 was found to be dominant in methyl orange (45%). In comparison with the bacterial community and azo dye structure, similar microbial phylum structures were observed in columns fed with methyl orange and methyl red, which have similar chemical structures. The appearance of *Firmicutes* in the columns with orange I and orange G was attributed to the naphthalene ring in their structures. Compared with the other columns, a higher abundance of *Proteobacteria* was reported in the columns with orange G, alizarin yellow R sodium salt and tartrazine, due to the presence of 2-sulfonic acid in their structures [28]. These results demonstrate the selective pressure behavior of azo

dyes on dewatered sludge, due to microbial diversity and adaptation. Simultaneously, the microorganisms with specific metabolic functions that showed more adaptability to the environment of dye wastewater, rapidly grew to be dominant populations [29]. This indicated that the flora from the dehydrated sludge had strong stress resistance and environmental adaptability.

### 3.3. Taxonomical and Functional Evaluation of the Acclimatized Sludge Based on T-RFLP Analysis

Genomic DNA was extracted from acclimatized sludge in seven different columns and used for T-RFLP analysis (Figure S3). The results of the T-RFLP analysis showed the unique bacterial community structures in each column after acclimatization. From the perspective of bacterial phyla, the original dewatered sludge contained 13 different types of bacteria (Figure 5). Among these, proteobacteria, bacteroidetes and firmicutes were observed to be the predominant phyla, with proportions of 35.9%, 17.7% and 17.6%, respectively, and the non-cultured phyla accounted for 13.2%. This analytical technique was performed for the comparative analysis of uncultured bacteria from acclimatized sludge with different azo dyes, as reported previously [30]. Additionally, the effect of uncultured bacteria on decolorization was also explored using different azo dyes.

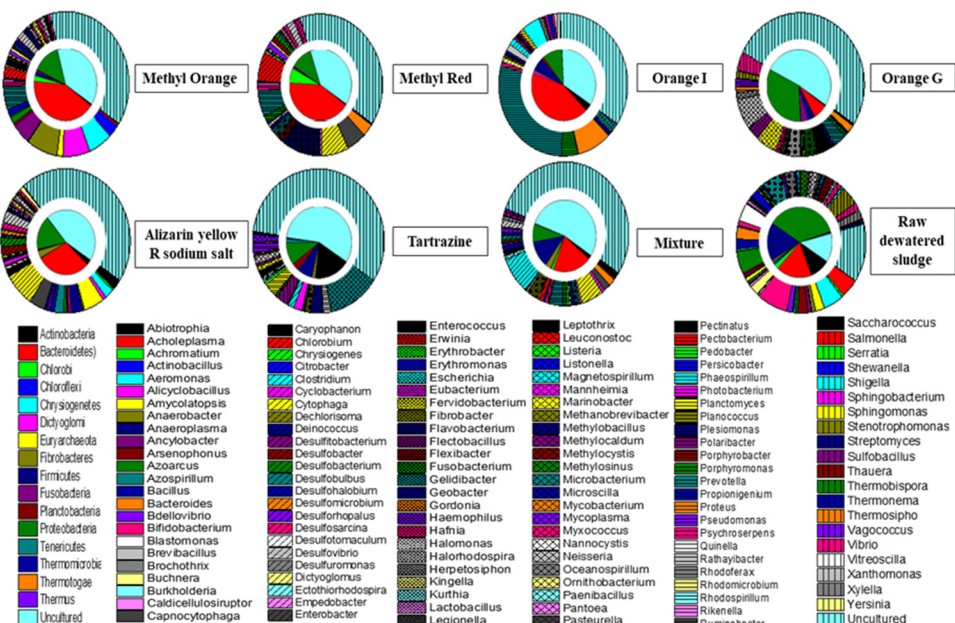

**Figure 5.** Double pie charts of microbial communities in activated sludge after acclimatization with different azo dyes. The inner pie represents the compositions of phyla, and the outer pie shows the compositions of genera. Above the pie is the azo dye used in each treatment system except the initial dewatered sludge. For better understanding, a color view is recommended.

It was clear that after acclimatization in the presence of different azo dyes, the microbial communities grew significantly [31]. Comparing initial dewatered sludge with the acclimatized sludge, the microbial community diversity was not only decreased but the decolorization of the azo dyes was also facilitated, consistently with previously reported studies [32]. The uncultured bacteria became the dominant strains and formed a large proportion of the microbial community. Meanwhile, the azo dyes acted as a recalcitrant compound to change the microbial habitat [33], resulting in mutated microorganisms adapted to the new environment and enhanced decolorization ability towards the azo dyes [34]. At the phylum level, uncultured bacteria and *Bacteroidetes* were the dominant phyla in the columns with methyl orange, methyl red, orange I and alizarin yellow R sodium salt, indicating a different microbial community from those with tartrazine and orange G. Additionally, the Proteobacteria were also found to be a dominant phylum in all these columns. These findings indicate that the Bacteroidetes can decolorize azo

dyes, particularly those with simple structures [35]. However, the decolorization of tartrazine and orange G, with relatively complex structures, was specifically attributed to the Actinobacteria and Proteobacteria phyla, respectively [36], while the Actinomycetes were regarded as the dominant bacteria for hydrazine decolorization [37]. Due to the different azo dye structures, the corresponding phylum structures and their abundances during the acclimatization process were different from each other. Furthermore, the abundances of the phyla in the mixed dyes column decreased to a certain extent, compared with the single dye columns. This can probably be attributed to the influence of the azo dye structures on the divergent succession of the bacterial community, reducing the diversity [38,39].

Genetic analysis further revealed that the uncultured strains formed high proportions of the genera of initial dewatered sludge and sludge with other treatments, thus playing an important role in decolorizing the azo dyes. Besides the uncultured strains, *Prevotella* (Bacteroidetes) and *Bacillus* (Firmicutes) were found to be the dominant genera in initial dewatered sludge with proportions of 7.8% and 5.1%, respectively. However, *Clostridium* (Firmicutes, 8.4%) and *Cytophaga* (Bacteroidetes, 5.5%) were regarded as dominant genera for the mixture of all dyes. Similarly, for methyl orange: *Flavobacterium* (Bacteroidetes, 7.9%), *Cytophaga* (Bacteroidetes, 7.0%), *Capnocytophaga* (Bacteroidetes, 6.1%) and *Prevotella* (Bacteroidetes, 5.4%); for methyl red: *Flavobacterium* (Bacteroidetes, 10.4%), *Cytophaga* (Bacteroidetes, 6.9%), *Chlorobium* (Chlorobi, 6.8%) and *Prevotella* (Bacteroidetes, 6.0%); for orange I: *Prevotella* (Bacteroidetes, 28.9%) and *Bacteroides* (Bacteroidetes, 8.5%); for orange G: *Neisseria* (Proteobacteria, 6.9%) and *Marinobacter* (Proteobacteria, 5.1%); for alizarin yellow R sodium salt: *Cytophaga* (Bacteroidetes, 7.7%) and *Flavobacterium* (Bacteroidetes, 5.1%); for tartrazine: *Microbacterium* (Actinobacteria, 12.7%) were observed to be dominant genera.

A significant genus shift was observed in comparison with initial dewatered sludge during the acclimatization process, and the obtained genus structures were also different from each other. These findings further confirmed that the differences in azo dye structures results in major microbial community changes. Moreover, some of the genera had low abundances but appeared in all columns. These mainly act as a medium for transferring electrons from microorganisms to the azo dyes in the system [40]. Due to these synergistic metabolic activities, the microbial community significantly enhanced the decolorization of the azo dyes [41]. In a microbial community, the individual strains may attack the dye molecule at different positions or may utilize metabolites produced by the co-existing strains for further decomposition [1].

### 3.4. CCA Analysis

In this analysis, a biplot correlation between the environmental variations and the microbial habitat (Figure 6) showed that all variables had same influence on microbial communities except D3 and CK. It was also evident from the biplot that CK and D3 were two independent habitats, differing significantly not only from all other dye treatments but also from each other. Further analysis revealed that D2 and D6, D1 and D7 were from similar territories but D4 and D5 had independent habitats. The relative importance of a particular parameter is illustrated by the length of the corresponding arrow line. The angles between the lines show the interrelationship of the variables, i.e., the closer the angles, the greater the correlation with each other [42]. The first and the second decolorization rates have the most extended lines (followed by initial pH), indicating a strong negative correlation with microbial habitats.

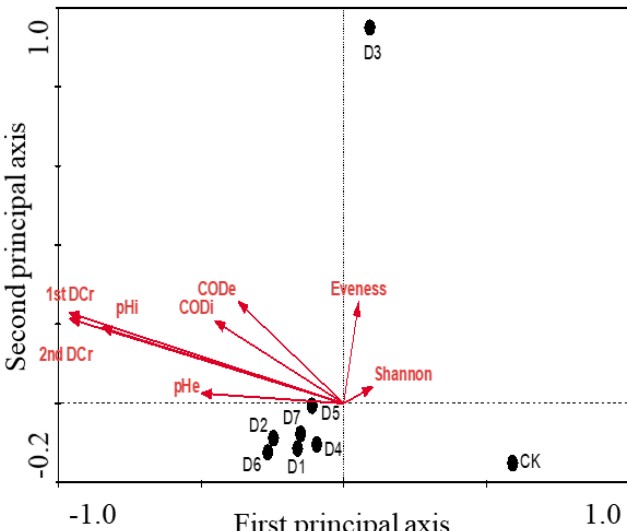

**Figure 6.** CCA analysis of habitat similarity of acclimatization systems with different azo dyes. DCr = decolorization rate, pHi = initial pH, pHe = final pH, CODi = initial COD, CODe = final COD, Shannon = Shannon diversity index, D1–D7 = the bacterial community structure of sludge acclimatized with methyl orange, methyl red, orange I, orange G, tartrazine, alizarin yellow R sodium salt and mixture and CK.

The relationship between the microbial distribution and microbial habitat is shown in Figure 7, where the microorganisms were divided into five groups (i.e., groups 1–5). Group 2 microorganisms mainly contributed to the habitat composition of CK, and the microorganisms in group 1 showed the highest diversity, highest relative abundance and a specific functional community for the decolorization of azo dyes (the specific diversity index is shown in Table S4). Group 3 showed a specific microbial community for the decolorization of tartrazine that was quite different from group 2 and group 1. However, group 4 and group 5 exhibited a transitional form of bacterial community, which played an important role in the formation of the microbial habitat. Except for tartrazine, the azo bonds in all the selected dyes were connected with the benzene ring, inducing the formation of conjugated double bonds and resulting in the further generation of aromatic intermediates having similar structures [43]. In tartrazine, one end of the azo band corresponds to the heterocycle (weakening the conjugation effect), inducing a significant change in the microbial community for reduction of the azo bond. In addition, aromatic compounds with a heterocyclic ring would further inhibit the degradation process, affecting the microbial community structure [44]. Hence, the azo dye structures can successfully determine the structural compositions of the bacterial communities. The transitional community also played a very important role in determining the microbial community succession and cannot be ignored, for example, in group 4 and group 5. Group 4 and group 5 played an important role in the successions of group 2 to 1 and group 1 to 3. Due to the unique structure of tartrazine among all the dyes, it is quite difficult for group 2 to be shifted to group 3 directly. Therefore, we have assumed the direction of the community succession was from group 2 to group 1 and from group 1 to group 3.

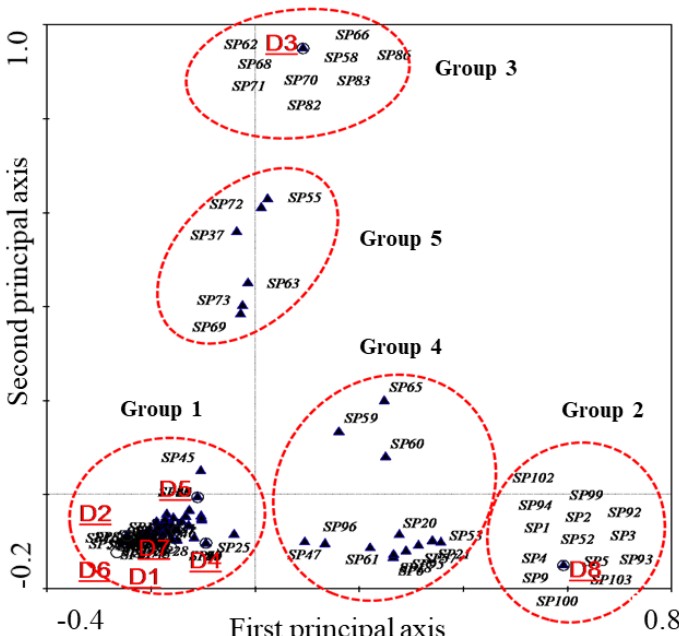

**Figure 7.** Biplot of species scores and habitats in a CCA of treatment systems acclimatized with different azo dyes. D1–D8 show the bacterial community structure of sludge acclimatized with various dyes (as mentioned above). Black triangles are presenting the specie and red circles were added to highlight different groups. Full names of species are presented in Table S5.

## 4. Conclusions

This study successfully investigated the microbial variations related to the chemical structures of different azo dyes and concluded that the chemical structure influenced decolorization and effective biodegradability. The microbial communities after acclimatization over different time periods with azo dyes underwent considerable succession. Compared with the activated sludge, the microbial diversity in the dewatered sludge decreased to some extent; however, an increased decolorization performance was observed. The presence of conjugated bonds and the bond energies of $-SO_3H$ and $-SO_2NH_2$ with azo rings further amplified the electron density. Therefore, the azo dyes with simple and lower molecular weight structures were found to be decolorized more easily and more effectively compared with those with complex structures. The chemical structures also influenced the bacterial community and caused a decrease in the Shannon diversity index, indicating the domination of specific microbial communities after domestication. It was also noticed that instead of a diversified microbial community, only the most common microbial groups were a major factor in the decolorization of azo dyes. Additionally, the possible succession route for microbial communities during the domestication process for dye decolorization was observed to be group 2–group 4–group 1–group 5–group 3. Dewatered activated sludge, being a natural source of nitrification bacteria, effectively degrades the azo dye wastewater. Furthermore, the structure of the azo dye can affect the composition of a highly efficient microbial community. This work provides a new insight into the design and operation of a high-performance bioreactor for treating textile wastewater containing azo dyes.

**Supplementary Materials:** The following are available online at https://www.mdpi.com/article/10.3390/w13213017/s1, Figure S1: Molecular identification; Figure S2: The concentration effect on the percentage (%) decolorization of methyl orange in the dewatered sludge; Figure S3: The metagenomic DNA gel electrophoresis patterns of different samples, Table S1: Chemical characteristics of inspected azo dyes; Table S2: Features of isolated strains from acclimated activated sludge; Table S3: Physiological-biochemical characteristics of isolated strains; Table S4: The diversity index of the bacterial communities of different treatments; Table S5: The full names of species.

**Author Contributions:** Conceptualization C.Z.; writing Z.M., M.S.S. and H.A.; review and editing H.W., M.S. All authors have read and agreed to the published version of the manuscript.

**Funding:** This work was financially supported by the Key Research and Development Plan of Shaanxi Province, (Program No. 2020NY-135).

**Institutional Review Board Statement:** Not applicable.

**Informed Consent Statement:** Not applicable.

**Data Availability Statement:** Not applicable.

**Conflicts of Interest:** The authors declare no conflict of interest.

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
