# Peer review of "Structure-Based Long-Term Biodegradation of the Azo Dye: Insights from the Bacterial Community Succession and Efficiency Comparison"

_water, doi:10.3390/w13213017_

Round 1

Reviewer 1 Report

Check attached file

Author Response

The following is a detailed response to the reviewers’ comments and suggestions.

Note: “Detailed Responses to Reviewers’ Comments” were marked with BLUE and all modifications and changes made in the revised files have been HIGHLIGHTED.

  1. The abstract should briefly state the biodegradation conditions of the tests in order to help the readers to centre the applicability of the conclusions. Note that the authors use dewatered sludge as primary source of microorganisms, which is not frequent. As the authors comment, this specific sludge could have some advantage since has suffered some selection because of the stress imposed by the dewatering process. This can limit the extrapolation of the conclusions to other systems starting from anaerobic sludge.

Response: Many thanks for your kind suggestion on enhancing the universality and guiding significance of our research! As suggested, we have made up the exact conditions of the biodegradation tests as “After 24-h acclimatization under room temperature without specific illumination, immediate de-colorization of the methyl red (89%) and methyl orange (78%) was observed because of their simple structure as compared to the tartrazine (73%).” and “Higher degrees of degradation and decolorization were achieved with Pseudomonas geniculate strain Ka38 (Proteobacteria), Bacillus cereus strain 1FFF (Firmicutes) and Klebsiella variicola strain RVEV3 (Proteobacteria) with continuous mixing at 30 ℃.”.

As the reviewer noted, dewatered sludge is a resource bank for various resistant functional bacteria in bio-treatment system. The dewatered sludge used in this study is derived from surplus biochemical sludge, that is, a mixture of aerobic and anaerobic activated sludge produced in the bio-treatment system of wastewater. Theoretically, it covers all nutrient-type functional microorganisms involved in the bio-treatment system, including aerobic, anaerobic and facultative anaerobic groups. Although after the dewatering and stacking process, the bacterial populations in the activated sludge will undergo succession, including the decline of some populations and the rise of some populations, at the same time, a large number of bacterial species have undergone physical and chemical characteristics mutations and tolerances increase based on environmental selection and horizontal gene transfer. This type of functional superposition will occur on aerobic, anaerobic and facultative anaerobic bacteria, so the isolation of functional bacteria and oxygen-sensitive bacteria types can be designed through subsequent domestication and separation methods (as in this study, we used the dewatered sludge as inoculum in the Winogradsky column with the azo dye solution to acclimate the functional groups in the  activated sludge and separate the facultative aerobic strains that can decolorize the azo dye). In summary, the use of dewatered sludge will not limit the development space of bacterial resources.

Please see modifications in the revised highlights.

  1. Although the introduction is very complete and instructive, the selection of references has not been the best possible. For instance, several references are devoted to aerobic processes (e.g., references 8 and 9), so they do not serve as basis for the present study. In addition, the more systematic approach to the use of anaerobic biofilms for azo-dyes biodegradation has been obviated ( Mezohegyi et al., Chem. Eng. J. 2008 143, 293-298; Mezohegyi et al., Ind, Eng, Chem, Res 2009, 48, 7054-7059; Mezohegyi et al., Appl. Catal. B: Env. 2010, 94, 179-185; Yu et al., J. Taiwan Inst. Chem. Eng. 2015, 54, 118-124; Balapure et al., J. Hazard. Mater. 2014, 279, 85-95).

Response: Thanks for recognizing the relevance of our work. We believe that this study could be a way forward for advancing the estimation of long-term biodegradation of the Azo Dye vs. bacterial community succession while treating the wastewater. According to the comments of the reviewer, a few relevant studies have been added in the revised manuscript as shown below:

In contrast with the conventional physico-chemical treatment methods for the wastewaters containing azo dye, previous studied also reported that the biological treatment approaches have gained much more popularity due to their low cost, less sludge production and environment friendly behavior. For instance, Mezohegyi et al. [8] studied anaerobic decolorization of azo dye Acid Orange in a continuous upflow stirred packed-bed reactor (USPBR) filled with biological activated carbon (BAC), and reported about 96% of bioconversion of azo dye in 0.5 min. Similarly, Balapure et al. [9] reported a complete decolorization and degradation of RB160 (100 mg/L) within 4 h along with co-metabolism of yeast extract (0.5%) by enriched mixed cultures BDN (BDN). The entry of azo dyes into the ecosystem also causes significant impact on microbial community. During the decolorization of azo dyes, the composition, activity and stability of microbial community structure play an important part. Yu et al. [10] evaluated the association in between decolorization and the microbial community structure and claimed about 85% and 75% of COD and methyl orange removal during the whole operation period, respectively.

  1. The experimental procedure is barely described, so the actual experimental design is hardly understood. A clear distinction must be done between each step, acclimatization or decolorization, mixed or not mixed dyes, and so on. 2/2

Response: Thank you for carefully pointing out our deficiencies in the method description. Here we reorganized our writing of materials and methods according to your advice. Firstly, we would like to clear the main technical route of this research by figure 1. The dewatered sludge was firstly domesticated by the Winogradsky column to obtain a microbial community with the function of degrading azo dyes and used to separate decolorizing functional strains, and then using the domesticated sludge and the separated functional strains to decolor single azo dyes and mixed azo dyes to compare their efficiency, and comparison and analysis of the microbial community structures in the sludge after the domestication of different azo dyes were operated at last.

Figure 1 Technical route of this study

The description order of the “material and method” was reorganized as follows:

  1. Materials and methods

2.1. Activated sludge

The sample of activated sludge was collected from the surplus sludge tanks of the full-scale municipal wastewater treatment plant (MWTP) located in Xi’an (N 34° 38' 20.89" E 109° 1' 14.47") [1718]. Sequential batch type intermittent feeding operation mode was adopted for sludge dewatering to further treat the different azo dyes [x].

2.2. Dyes and reagents

Methyl Orange, Methyl Red, Orange I, Orange G, Tartrazine and Alizarin yellow R sodium salt were selected as the model azo dyes because of their great use in textile industry, and were purchased from Sino Pharm Chemical Reagent Co., Ltd (Table S1). The biochemical reagents were procured from Beijing CW Biotech Co., Ltd. All other reagents used in this study were of analytical grade.

2.3. Sludge acclimatization

Acclimatization experiments were performed in the Winogradsky column with the height, diameter, and thickness of 99.0 cm, 5 cm, and 1 cm, respectively (Figure 2). The dewatered sludge (100 g) was inoculated in acclimatized medium (800 mL) containing 30 mg/L of single azo dye each in six differ columns and 1 column with mixture of 6 azo dyes mentioned above under room temperature without specific illumination. Selective culture medium contained Na2SO4 (5.0 g/L), CaCO3 (2.5 g/L), microcrystalline cellulose (7.0 g/L), peptone (10.0 g/L), and NaCl (5.0 g/L) at neutral pH. Yeast extract of about 10.0 g/L was used as nutrient. After complete decolorization, the supernatant was removed, and new culture medium and dye was added for the next acclimatization cycle. At predetermined time intervals, 2 mL samples were taken, filtered, centrifuged and the dye concertation was spectrophotometrically determined (UV2300 spectrophotometer China).

2.4. Strain isolation

The About 10 mL of domesticated sludge from Winogradsky column was trans-ferred placed into 90 mL of sterile water (containing glass beads) followed by rapid mixing (10 min), to obtain the solution with the dilution concentration of 10-1. Then 1 mL of bacterial liquid (Conc: 10-1) in 9 mL of sterile water was placed and mixed well to obtain 10-2 concentration of bacterial liquid. The same method was adopted to prepare the dilution till 10-8. The 0.5 mL of bacterial solution sampled from 10-6, 10-7, 10-8 dilution test tubes was applied to solid separation medium and incubated for 24 h at 35 ℃. Some of the colony photos are shown in Figure 1. Different forms of single colonies were selected to inoculate into liquid separation medium at 30 ℃ for 12 -hr to extend culturing and purification. In end, the cultured bacteria liquid was placed on the solid separation medium for dashing operation to obtain pure species single colony, for dye culture medium decolorization experiments. Total 9 strains were finally isolated and coded as T-1, T-2, T-3, T-4, T-5, T-6, T-7, T-8, T-9, and their morphological and the physiological characteristics are presented in supplementary information (Table S2 & S3).

2.5. Strain identification

The morphological, physiological, and biochemical characteristics of the strains were determined with reference to the previously reported method by [1819]. Genomic DNA was extracted from acclimatized sludge using a SoilGen DNA Kit followed by the manufacturer’s protocol. However, the purified DNA of the strains was used as a template for the polymerase chain reaction (PCR) amplification with 16S rRNA gene using bacterial community. The PCR amplifications were performed by 27F and 1492R as the forward and reverse primer, respectively. The PCR mixtures (50 µL) contained 10 × PCR buffer 5 µL, dNTP (2.5 mM) 3.2 µL, rTaq (5 U/µL) 0.4 µL, Fam-27F (5 mM) 2 µL, and genomic DNA (50 ng) added into the 50 µL of ddH2O. The following PCR program was used on T-gradient (Biometra Germany); 10 mins 94 ℃, 30 s 94 ℃, 30 s 55 ℃, 45 s 72 ℃, 30 cycles and a final extension of 10 mins 72 ℃. PCR products were then visualized on 1.8% of agarose gels strained with ethidium bromide. After detecting the size of the amplified product by agarose gel electrophoresis, TA (thymine and adenine) cloning and sequencing was performed. Subsequently, the results of the sequenced 16S rRNA genes were compared with the related 16S rRNA gene sequence on GenBank. For similarity comparison analysis, ClustalX and Mega5.1 software were used to construct the phylogenetic tree (Fig. S1).

2.6. Decolorization test of isolated strains

The isolated colonies were inoculated into the 30 mL of liquid medium under aseptic operating conditions and cultured for 12 hours with smooth mixing till the OD600 was about 0.6). After culturing, 3 mL cultured solution of each colony was separately inoculated into 150 mL dye medium with continuous shaking at 30 ℃. Subsequently, 2 mL of inoculated suspension was collected and centrifuged at 10000 r/min for 5 minutes. The collected samples (1 mL) were then utilized to determine the UV-Vis absorbance at the maximum absorption wavelength of the azo dye. UV-Vis wavelength scanning was registered within the range of 200-800 nm. The decolorization rate was calculated by using the following equation:

Decolorization (%) =            (1)

Where A(t0) and A(t) are the initial Absorbance intensity (i.e., time = 0 h) and the absorbance intensity after a particular reaction time (i.e., time = t), respectively. However, the effect of mixed inoculated dyes in one solution was also observed, separately. All isolated strains were cultured in 30 mL of liquid medium at 30 ℃ with continuous mixing for 12 hours. The mixed dyes solution contains methyl orange, golden orange I, orange yellow G, hydrazine yellow, and alizarin R sodium salt, with the concentration of 30 mg/L for each dye.

2.7. T-RFLP analysis of acclimated sludge

Terminal-restriction fragment length polymorphism (T-RFLP) analysis was adopted to reveal and compare the bacterial communities of all acclimated sludge and raw dewatered sludge. Genetic profiles of the amplified bacterial 16S rRNA were generated by restricted enzyme digestion. Soil DNA kit (Beijing Kangwei Century Biotechnology Co., Ltd.) was utilized for the DNA extraction from acclimatized activated sludge with different azo dyes. The reaction mixture (25 μm) contains 1 μm of each purified product as template, 12.5 μL 2 × Taq Plus PCR Master-Mix mixture, 9.5 μm of ddH2O and 1 μm of reaction buffer (27F/1492R). The reaction was carried out at 95 ℃ - 5 mins, 94 ℃ - 1 min, 60 ℃ - 45 sec, and 72 ℃ - 1 min for 25 cycles. Size separation of terminal restriction fragments was performed by capillary gel electrophoresis using an ABI gene analyzer 3130XL (Biosystems USA) equipped with capillaries loaded by POP4 polymer (Thermo fisher, USA). Relative migration of restricted fragments was determined by GeneMarker (HID V1.7). Finally, by removing primer peaks (>50 bps) and spurious peaks (relative abundance > 0.5 %), the T-RFLP profile was figured out. The relative peak area of a single T-RF was calculated according to the following equation:

Ap = Ni/N×100                                                                                                             (2)

Where the Ni and N are the peak areas of the single and total T-RFs, respectively. T-RFs were analyzed by MiCA3 PAT and referred to the website (http://mica.ibest.uidaho.edu/pat.php), from which bacterial community structure spectrums were constructed.

2.8. CCA analysis:

Correlation analysis was conducted on species composition and variables factors by using canonical correspondence analysis (CCA) with CANOCO (Windows, 4.5 package). Parameters like COD, pH, decolorization rate, evenness and Shannon diversity index was selected as variable factors for the seven different samples that were taken from reactor. The CCA biplot characterize the biological communities relative to the selected variables. The arrows of the biplot represents the variable factors, the length of arrow represent the maximal value at its tip [1920].

  1. The decolorization method selected does not seem the best option according to the most recent publications that prefer purely anaerobic environment where the decolorization is very effective. The authors affirm that operation progressed under continuous mixing, but how this mixing was conducted? How then two section of aerobic and anaerobic conditions were maintained?

Response: Thanks to the reviewer for pointing out the inaccuracy in the description of the decolorization method in the research article. The "mixing" that appears here is exactly to realize the full contact and mixing of the dye molecules and the strain culture through shaking of the shaker, in order to obtain the ideal decolorization and degradation effect. The shaking and mixing here simulates mechanical stirring and mixing or aeration and mixing in the process of sewage biochemical treatment. In terms of method description, the most accurate statement is continuous “shaking”. We have revised the original text to accurately describe the facts.

In the process of shaking and mixing, the dissolved oxygen in the upper layer of the culture vessel is close to the aerobic level, while in the lower layer of the solution is still close to the anaerobic level even when the shaking is enhancing re-oxygenation. In a sense, a facultative aerobic environment is realized, which is suitable for the metabolic characteristics of the isolated strains in this study, so as to achieve higher decolorization efficiency.

The biological anaerobic method is indeed more efficient in the decolorization of dye wastewater. With the breakage of the chemical bonds of the dye macromolecules and the destruction of the color group during the acidification process, the color of the wastewater can be removed, but the anaerobic treatment cannot completely degrade the dye. Therefore, aerobic treatment is the prevalent choice for subsequent treatment [1,2]. The strains isolated in this study are all facultative aerobic bacteria (identified by the physical and chemical properties), which can live under aerobic or anaerobic conditions. When there is oxygen, they can pass the respiratory energy and anaerobic. It can also produce energy through fermentation or anaerobic respiration. Whether under aerobic or anaerobic condition, the molecular structure of the dye could be destroyed by microbial oxidation or fermentation to achieve decolorization and degradation.

  1. Kiran, S., Ali, S., & Asgher, M. (2013). Degradation and mineralization of azo dye reactive blue 222 by sequential photo-Fenton’s oxidation followed by aerobic biological treatment using white rot fungi. Bulletin of environmental contamination and toxicology90(2), 208-215.
  2. Aleboyeh, A., Olya, M. E., & Aleboyeh, H. (2008). Electrical energy determination for an azo dye decolorization and mineralization by UV/H2O2 advanced oxidation process. Chemical Engineering Journal137(3), 518-524.
  3. Always put one space between the values of some variable and its units. Now, this has not been made everywhere.

Response: Thank you for the suggestion, the whole manuscript has been revised again carefully, to ensure the space slot in between the variables and their units.

  1. In section 3.1, what means “three groups according to the time interval 201 (1: 8: 8 days)”? What expresses this classification? In addition, In Figure 3, the colours are just labelled as “first”, “second” and “third” (???).

Response: Thank you for the comment. Ratio of 1:8:8 represent the time interval of sampling to measure the acclimatization stage. For that purpose, sampling was divided into three groups on the basis of time, 1st group samples were taken right after 24 hrs (1 day), however, 2nd and 3rd group sampling was performed after 8 days. Here, the samples of 3rd group were take only to detect the acclimatization based absorbance removal. Additionally, Figure 3 represent the decolorization of these three groups, i.e. First Group (1 day), Second Group (8 day) Third Group (8 day). According to the reviewer’s suggestion changes have been incorporated in revised manuscript (Color Red), and are also given below:

  The first group of samples was taken after 24-hr, and the samples of the second and third groups were taken after eight days. The acclimatization of third group was performed to analyze the absorbance.

  1. In section 3.4, DCr should be DC since the parameter used is the decolorization in %, not a rate.

Response: Thank you for the constructive suggestion, recommendation has been incorporated in the revised manuscript.

  1. Please always keep a space between the value and its units. Too many times, this rule is not followed.

Response: Apologies for mishandling the typing mistake and thank you so much for the highlighting the errors. The whole manuscript has been revised again carefully, to ensure the space slot in between the variables and their units.

  1. Line 22. The symbol for hour is h, not hr. It appears in other parts of the manuscript, although h is also used. Please, correct it everywhere.

Response: Thank you for the comment. Suggested recommendation has been incorporated in the revised manuscript.

  1. In fact, a carefully revision of units and symbols should be performed as there are frequent errors. See, for instance, lines 178-179, “The reaction was carried out at 95 ℃-5mins, 94 ℃-1min, 60 ℃-45sec, 178 and 72 ℃-1min for 25 cycles” (no space between time and units, mins is min and sec is s).

Response: Thank you for the highlighting the errors in manuscript. The whole manuscript has been revised carefully as per the reviewer’s suggestion, to ensure the space in between the variables and their units.

  1. In Figure S2, the y-axis represents decolorization as expressed in equation 1, not decolorization rate.

Response: Thank you for the comment. Suggested modification has been incorporated in Figure S2.

  1. Line 207, “ascribed”.

Response: Thank you for the highlighting the typing error, suggestion has ben incorporated in revised manuscript.

  1. Line 222 and line 235, “was”.

Response: Thank you for your critical analysis and highlighting the grammatical mistakes.  As According to the reviewer’s suggestion, changes have been incorporated in revised manuscript, along with complete and careful revision or whole manuscript.

  1. In general, it is recommended a revision by an English-speaking native since frequent incorrect expressions are found, mainly in the Discussion section.

Response: Thank you for the comment and valuable suggestion. Whole manuscript has been revised again, by specifically keeping in mind the incorrect expressions in Discussion sections.

Reviewer 2 Report

I believe there are some interesting findings here but the Methods section requires major revisions as the current description of the work is confusing. The Results/Discussion section also is in need of significant improvement. Specific comments are provided below for each section. There were many small grammar/typo (English-language) issues that I have not included (a copy editor should catch those).

Abstract

line 26-28: not clear how this conclusion was reached, but my understanding of experimental design may be incomplete (see comments in Methods below)

l. 30: probably true, but no replicates so could be stochastic (see comments below)

Introduction

l. 53: “population” should be “assemblage” or “community”

l. 52-54: awkward sentence

l. 56: is a specific “biological process,” e.g., dye degradation, meant here?

l. 59: not clear what is meant here by “behavior”

l. 66: of the targeted dye? or associated with the microbes themselves? Please clarify

l. 80-82: Confusing sentence. Unclear what “mechanism of the microbial community” means and community composition is part of community structure so wording here should be tweaked

l. 86: needs citations here

l. 87: change “flora” to “microbes” (or similar word)

l. 90: give citations to go along with “few,” or say “none” here

l. 99: wording here could be clearer…isn’t the microbial habitat the sum of the various factors in the environment?

Methods

l. 106: need citations here

l. 112: isolating strains is not an experiment; remove “experiment” from subheading

l. 112: Overall, I’m confused as to when these colonies were isolated and how used. I think they were isolated from your Winogradsky columns after incubation with dyes. If so, this section should be moved to later in the Methods section. If not, it sounds like they were introduced into those Winogradsky columns, along with the complete bacterial assemblages already in the sludge. Such an experimental design would be confusing, and if that is the case, better description of, and explanation for, is needed.

l. 117: not clear what a “bacteria liquid coated sample” is

l. 120: “Different forms of single colonies were selected” This seems to imply that not all colonies were picked. Therefore, you’re getting a very biased subset of the community, not only because you’re missing all non-culturable taxa, but also picking a subset (by eyeballing different- (or similar-) looking colonies). Wording later in the manuscript makes it sound like these were the only strains (or at least the only culturable ones), but if only a subset was picked, this is in doubt

l. 122: “dashing operation” … does this mean streak plating?

Table S2: This table makes it appear that your 9 isolates matched the GenBank strains exactly (it’s later made clear that this is not the case). You should label the GenBank strains as something like “Closest Relative” and give the % similarities in the table. Also, some of your descriptors (“Big” and “Thick”) should be better defined, e.g., with numerical ranges. Also, not clear what “Color difference” means and “Water solubility” seems to refer to color?

Table S3: “Urease”?

l. 128 (Fig. 1 legend): legend description needed for right-hand photo

l. 131: Sounds like you extracted sludge DNA but didn’t use it…but later it’s clear that you did sequence from this extracted DNA. Clarify wording here.

Fig. S1: Describe the gel image in figure legend or omit. Tree is probably sufficient here

l. 147: It appears that each treatment (different dye in columns) was unreplicated. Clarify if this is incorrect. This makes some of your interpretations weak (see Results/Discussion comments)

l. 154: What does the 1 cm thickness here refer to?

l. 154: I’m confused by this sentence. It sounds like your 9 isolates were added in addition to the bacteria in the sludge to the Winogradsky columns. But, if so, which isolates to which columns (dyes or dye combos)? Later in the manuscript it sounds like this was not the case, and instead the 9 strains were isolated from the bacterial assemblages in these columns after changes (“succession”) during the incubation of these columns. But I’m not 100% sure which it is.

Fig. 2: Need more description in legend. What are the 8 dye solutions (I count 7 from text)? What is the rainbow showing – light penetration?

l. 173: “restriction” enzyme. I’m not clear why the T-RFLP work was done. Much more information was derived from the DNA sequencing (if I’m understanding the Methods description correctly). The T-RFLP is used to show assemblages in the colonies were “different” but that’s also clear from the CCA of sequencing data

l. 176: This sounds like each reaction mixture contained multiple templates. I believe you mean microliters (ul) here, not micrometers (um). The volumes don’t add up to 25 ul.

l. 196-197: This sentence could instead go in to the Fig. 6 legend

Results & Discussion

l . 202: I’m confused by the “1: 8: 8 days” These apparently refer to sampling times at 1 day, 8 days, and ???

Fig. 3: What are the “three circles”? Does this refer to the 3 sampling times? Also, “Miture” to “Mixture”

l. 220: What is “smooth” decolorization?

l. 235: Is this really a “rate” or rather a % or “efficiency”?

l. 237: Wording could be clearer: “resulted in lower biomass production” More importantly, what is the evidence for this?

Fig. 4: Are these proportions based on (number of colonies of strain X)/(totaI number of colonies from a Winogradsky column)? If so, make this clearer in Methods section. I believe the results here are based in single replicates (a single column), but correct this in Methods if this is incorrect.

l. 249, 252: some of your rank orders are not exactly correct

l. 254-255: Unsure what is meant by “Comprehensive physiological and morphological analysis further showed that the isolates obtained in this study were mainly Bacillus sp., Stenotrophomonas sp., Paracoccus sp., and Pseudomonas sp.” I thought the colonies were identified by DNA sequencing. Would the physiological and morphological analyses change your identifications?

l. 256-261: Although both of these (phenotypic changes, evolutionary adaptation) may have occurred, this study only shows changes in bacterial assemblage structure during incubation.

l. 265: This should be in Table S2 and made clear that these GenBank taxa are just the closest sequence matches, rather than the very same strains

l. 268: should be “Bacillus anthracis strain ####”

l. 282: some of these likely would not be new species, esp. the ones with very high (99%) sequence similarity

l. 288-289: these are not phyla

l. 290: not the most dominant in all 4 of these treatments

l. 292-297: Where are these %s coming from? Don’t seem to agree with those in Fig. 4.

l. 297: Meaning of this sentence is unclear: “In comparision with bacterial community and azo dye structure, similar phyla composition was observed both in columns and in the structures of Methyl Orange and Methyl Red.”

l. 313: text says 13 phyla, figure legend implies 16

l. 335: Is “Actinobacteria” here a typo? Are you trying to say that these two dye treatments were different, especially in that they had higher %s of Actinobacteria?

l. 343: most of this paragraph is redundant with the prior paragraph

l. 349: “Hydrazine” comes out of nowhere. Citation needed?

l. 350: this statement (and many others like it) cannot be supported statistically because no replicates. It’s possible that stochastic factors, rather than the dye in each column, were the cause of differences in diversity, community structure, etc. At a minimum, this limitation of the study must be mentioned in the Discussion.

l. 380-386: redundant with Methods (or should be moved there)

l. 384: Densities not actually measured. Perhaps “relative abundance” would be OK here

Fig. 6: What are D-1 through D-7? Define each one in legend. Change “principle” to “principal”

It’s odd that your variables here are a mix microcosm conditions, bacterial assemblage characteristics, and effects on dyes, i.e., a mix of causes and effects

l. 388: “all variables have same influence on microbial communities” See comment above about cause and effect

l. 390: “differ not only from all other variables” I think the intent here is to say “differ not only from all other dye treatments”

l. 391: what is meant here by “in-depth analysis”?

l. 394: “centrifugal”?

l. 406: by “density” do you mean the most species?

l. 409: these are not isolates

l. 414: “followed the same mechanism” Identifying the mechanisms for decolorization was a stated goal of the study. Although you may have identified some of the microbial players in the process, I don’t think you can say much about the actual mechanisms

l. 420: “composition of the community structure” Composition is part of the structure, so this should be re-worded

l. 422: As far as I can tell, the data in Fig. 7 come from a single sampling time so really can’t say anything about succession. However, this may be incorrect because some sampling was done at three time points. In any case, it should be made clear in the Methods section from which time point(s) the data for Figs. 4-7 are derived.

l. 424: no evidence for this statement

Fig. 7: CK is now D8?

l. 429: No appendix?

Conclusions

l. 431: “investigated the microbial dynamics of the azo dyes having different chemical structures” Wording here could be clearer

l. 440: “contribution rate of Shannon diversity index” Awkward phrasing

l. 422: meaning unclear

l. 443-445: this speculation should be removed (see comment above)

Author Response

We highly appreciate your comments to make this manuscript better

Abstract

line 26-28: not clear how this conclusion was reached, but my understanding of experimental design may be incomplete (see comments in Methods below)

Response: Many thanks for your kind suggestion on enhancing the universality and guiding significance of our research! As suggested, we have made up the exact conditions of the biodegradation tests as “After 24-h acclimatization under room temperature without specific illumination, immediate de-colorization of the methyl red (89%) and methyl orange (78%) was observed because of their simple structure as compared to the tartrazine (73%).” and “Higher degrees of degradation and decolorization were achieved with Pseudomonas geniculate strain Ka38 (Proteobacteria), Bacillus cereus strain 1FFF (Firmicutes) and Klebsiella variicola strain RVEV3 (Proteobacteria) with continuous mixing at 30 ℃.”.

As noted, dewatered sludge is a resource bank for various resistant functional bacteria in bio-treatment system. The dewatered sludge used in this study is derived from surplus biochemical sludge, that is, a mixture of aerobic and anaerobic activated sludge produced in the bio-treatment system of wastewater. Theoretically, it covers all nutrient-type functional microorganisms involved in the bio-treatment system, including aerobic, anaerobic and facultative anaerobic groups. Although after the dewatering and stacking process, the bacterial populations in the activated sludge will undergo succession, including the decline of some populations and the rise of some populations, at the same time, a large number of bacterial species have undergone physical and chemical characteristics mutations and tolerances increase based on environmental selection and horizontal gene transfer. This type of functional superposition will occur on aerobic, anaerobic and facultative anaerobic bacteria, so the isolation of functional bacteria and oxygen-sensitive bacteria types can be designed through subsequent domestication and separation methods (as in this study, we used the dewatered sludge as inoculum in the Winogradsky column with the azo dye solution to acclimate the functional groups in the  activated sludge and separate the facultative aerobic strains that can decolorize the azo dye). In summary, the use of dewatered sludge will not limit the development space of bacterial resources.

Please see modifications in the revised highlights.

  1. 30: probably true, but no replicates so could be stochastic (see comments below)

 Response: Thank you for the constructive suggestion, recommendation has been incorporated in the revised abstract. The lack of replicate settings does weaken the original conclusion that the bacterial community structure in the sludge depends on the chemical structure of the dye molecules used for domestication. Even if there are repeated settings, the original conclusion is still too confirmative in the presence of existing data, therefor we have weakened the affirmative statement, only pointed out this phenomenon and possible reasons. The original conclusion of “Moreover, the formation of the microbial community was determined by azo dyes structure, but these structures show negligible effect on the diversity of microbial community.” was revised to “Moreover, it seems that the chemical structures of azo dyes, in a sense, drove the divergent succession of the bacterial community while lowering the diversity.”

Introduction

  1. 53: “population” should be “assemblage” or “community”

Response: Thank you for the constructive suggestion, recommendation has been incorporated in the revised manuscript.

  1. 52-54: awkward sentence

Response: Thanks for the comment and suggestion regarding content of the manuscript, highlighted sentenced has been rephrased in the revised manuscript as per the reviewer’s demand.

  1. 56: is a specific “biological process,” e.g., dye degradation, meant here?

Response: Thanks for the comment. Yes, here the term “biological process” specifically means to say the dye degradation. Moreover, suggestion has been incorporated in the revised manuscript.

  1. 59: not clear what is meant here by “behavior”

Response: Thanks for the nice suggestion to increase the worth of our manuscript. Here, author trying to say the removal behavior of dyes, apologies for this inconvenience. Highlighted error has been modified as “removal behaviors” in the revised manuscript.

  1. 66: of the targeted dye? or associated with the microbes themselves? Please clarify

Response: Thanks for the comment. The expression here is not complete enough. We want to say that clarifying the microbial community structure in different azo dye degradation systems is of great influence on the biological treatment performance of azo dyes. So, the modification should be as “Therefore, the microbial community structure and diversity may significantly affect the performance and stability of the biological processes of azo dyes (e.g., dye degradation) as reported elsewhere.”

  1. 80-82: Confusing sentence. Unclear what “mechanism of the microbial community” means and community composition is part of community structure so wording here should be tweaked

Response: Thank you for the recommendation and carefully reviewing the manuscript. Suggested recommendation has been incorporated in the revised manuscript as shown below:

Hence, the studies regarding the response of the microbial community against the degradation of azo dyes are needed to be explored on the basis of microbial community structure, to further enhance the performance efficiency.

  1. 86: needs citations here

Response: Thank you for the comment. Citation has been added in the revised manuscript.

  1. 87: change “flora” to “microbes” (or similar word)

Response: Thank you for the constructive suggestion, recommendation has been incorporated in the revised manuscript.

  1. 90: give citations to go along with “few,” or say “none” here

Thank you for the comment, suggested recommendation has been incorporated in the revised manuscript.

  1. 99: wording here could be clearer…isn’t the microbial habitat the sum of the various factors in the environment?

 Response: Thanks for the query. It is true that the microbial habitat is actually the sum of all factors while our statement here is not thoughtful, so we have deleted the redundant description “In addition, polyacrylamide flocculants are added to the sludge during the dewatering process, so that the microbes in the dewatered sludge may have the potential to de-grade organic.”

Methods

  1. 106: need citations here

Response: Thank you for the comment. Citation has been incorporated in the revised manuscript.

  1. 112: isolating strains is not an experiment; remove “experiment” from subheading

Response: Thank you for the valuable suggestion. Suggested recommendation has been incorporated in the revised manuscript.

  1. 112: Overall, I’m confused as to when these colonies were isolated and how used. I think they were isolated from your Winogradsky columns after incubation with dyes. If so, this section should be moved to later in the Methods section. If not, it sounds like they were introduced into those Winogradsky columns, along with the complete bacterial assemblages already in the sludge. Such an experimental design would be confusing, and if that is the case, better description of, and explanation for, is needed.

Response: We are very sorry that our description in the part of experimental materials and methods is not precise enough and the order is not clear. We have readjusted the description in this part. Firstly, we would like to clear the main technical route of this research by figure 1. The dewatered sludge was firstly domesticated by the Winogradsky column to obtain a microbial community with the function of degrading azo dyes and used to separate decolorizing functional strains, and then using the domesticated sludge and the separated functional strains to decolor single azo dyes and mixed azo dyes to compare their efficiency, and comparison and analysis of the microbial community structures in the sludge after the domestication of different azo dyes were also operated. Finally, we tried to establish certain relationship between the chemical structure of the tested azo dyes and the bacterial communities of the acclimated sludge. Combining the information of the bacterial community in the activated sludge after the domestication of different dyes, we found that there seems to be a certain relationship between the structure of the dye and the succession of the bacterial community, which is a kind of selective pressure or external cause driving the divergent succession of the microbial community of the raw sludge.   

Figure 1 Technical route of this study

The description order of the “material and method” was reorganized as follows:

  1. Materials and methods

2.1. Activated sludge

The sample of activated sludge was collected from the surplus sludge tanks of the full-scale municipal wastewater treatment plant (MWTP) located in Xi’an (N 34° 38' 20.89" E 109° 1' 14.47") [1718]. Sequential batch type intermittent feeding operation mode was adopted for sludge dewatering to further treat the different azo dyes [x].

2.2. Dyes and reagents

Methyl Orange, Methyl Red, Orange I, Orange G, Tartrazine and Alizarin yellow R sodium salt were selected as the model azo dyes because of their great use in textile industry, and were purchased from Sino Pharm Chemical Reagent Co., Ltd (Table S1). The biochemical reagents were procured from Beijing CW Biotech Co., Ltd. All other reagents used in this study were of analytical grade.

2.3. Sludge acclimatization

Acclimatization experiments were performed in the Winogradsky column with the height, diameter, and thickness of 99.0 cm, 5 cm, and 1 cm, respectively (Figure 2). The dewatered sludge (100 g) was inoculated in acclimatized medium (800 mL) containing 30 mg/L of single azo dye each in six differ columns and 1 column with mixture of 6 azo dyes mentioned above under room temperature without specific illumination. Selective culture medium contained Na2SO4 (5.0 g/L), CaCO3 (2.5 g/L), microcrystalline cellulose (7.0 g/L), peptone (10.0 g/L), and NaCl (5.0 g/L) at neutral pH. Yeast extract of about 10.0 g/L was used as nutrient. After complete decolorization, the supernatant was removed, and new culture medium and dye was added for the next acclimatization cycle. At predetermined time intervals, 2 mL samples were taken, filtered, centrifuged and the dye concetration was spectrophtometrically determined (UV2300 spectrophotometer China) .

2.4. Strain isolation

The About 10 mL of domesticated sludge from Winogradsky column was trans-ferred placed into 90 mL of sterile water (containing glass beads) followed by rapid mixing (10 min), to obtain the solution with the dilution concentration of 10-1. Then 1 mL of bacterial liquid (Conc: 10-1) in 9 mL of sterile water was placed and mixed well to obtain 10-2 concentration of bacterial liquid. The same method was adopted to prepare the dilution till 10-8. The 0.5 mL of bacteria liquid coated sample from 10-6, 10-7, 10-8 dilution test tubes was applied to solid separation medium and incubated for 24 h at 35 ℃. Some of the colony photos are shown in Figure 1. Different forms of single colonies were selected to inoculate into liquid separation medium at 30 ℃ for 12 -hr to extend culturing and purification. In end, the cultured bacteria liquid was placed on the solid separation medium for dashing operation to obtain pure species single colony, for dye culture medium decolorization experiments. Total 9 strains were finally isolated and coded as T-1, T-2, T-3, T-4, T-5, T-6, T-7, T-8, T-9, and their morphological and the physiological characteristics are presented in supplementary information (Table S2 & S3).

2.5. Strain identification

The morphological, physiological, and biochemical characteristics of the strains were determined with reference to the previously reported method by [1819]. Genomic DNA was extracted from acclimatized sludge using a SoilGen DNA Kit followed by the manufacturer’s protocol. However, the purified DNA of the strains was used as a template for the polymerase chain reaction (PCR) amplification with 16S rRNA gene using bacterial community. The PCR amplifications were performed by 27F and 1492R as the forward and reverse primer, respectively. The PCR mixtures (50 µL) contained 10 × PCR buffer 5 µL, dNTP (2.5 mM) 3.2 µL, rTaq (5 U/µL) 0.4 µL, Fam-27F (5 mM) 2 µL, and genomic DNA (50 ng) added into the 50 µL of ddH2O. The following PCR program was used on T-gradient (Biometra Germany); 10 mins 94 ℃, 30 s 94 ℃, 30 s 55 ℃, 45 s 72 ℃, 30 cycles and a final extension of 10 mins 72 ℃. PCR products were then visualized on 1.8% of agarose gels strained with ethidium bromide. After detecting the size of the amplified product by agarose gel electrophoresis, TA (thymine and adenine) cloning and sequencing was performed. Subsequently, the results of the sequenced 16S rRNA genes were compared with the related 16S rRNA gene sequence on GenBank. For similarity comparison analysis, ClustalX and Mega5.1 software were used to construct the phylogenetic tree (Fig. S1).

2.6. Decolorization test of isolated strains

The isolated colonies were inoculated into the 30 mL of liquid medium under aseptic operating conditions and cultured for 12 hours with smooth mixing till the OD600 was about 0.6). After culturing, 3 mL cultured solution of each colony was separately inoculated into 150 mL dye medium with continuous shaking at 30 ℃. Subsequently, 2 mL of inoculated suspension was collected and centrifuged at 10000 r/min for 5 minutes. The collected samples (1 mL) were then utilized to determine the UV-Vis absorbance at the maximum absorption wavelength of the azo dye. UV-Vis wavelength scanning was registered within the range of 200-800 nm. The decolorization rate was calculated by using the following equation:

Decolorization (%) =            (1)

Where A(t0) and A(t) are the initial Absorbance intensity (i.e., time = 0 h) and the absorbance intensity after a particular reaction time (i.e., time = t), respectively. However, the effect of mixed inoculated dyes in one solution was also observed, separately. All isolated strains were cultured in 30 mL of liquid medium at 30 ℃ with continuous mixing for 12 hours. The mixed dyes solution contains methyl orange, golden orange I, orange yellow G, hydrazine yellow, and alizarin R sodium salt, with the concentration of 30 mg/L for each dye.

2.7. T-RFLP analysis of acclimated sludge

Terminal-restriction fragment length polymorphism (T-RFLP) analysis was adopted to reveal and compare the bacterial communities of all acclimated sludge and raw dewatered sludge. Genetic profiles of the amplified bacterial 16S rRNA were generated by restricted enzyme digestion. Soil DNA kit (Beijing Kangwei Century Biotechnology Co., Ltd.) was utilized for the DNA extraction from acclimatized activated sludge with different azo dyes. The reaction mixture (25 μm) contains 1 μm of each purified product as template, 12.5 μL 2 × Taq Plus PCR Master-Mix mixture, 9.5 μm of ddH2O and 1 μm of reaction buffer (27F/1492R). The reaction was carried out at 95 ℃ - 5 mins, 94 ℃ - 1 min, 60 ℃ - 45 sec, and 72 ℃ - 1 min for 25 cycles. Size separation of terminal restriction fragments was performed by capillary gel electrophoresis using an ABI gene analyzer 3130XL (Biosystems USA) equipped with capillaries loaded by POP4 polymer (Thermo fisher, USA). Relative migration of restricted fragments was determined by GeneMarker (HID V1.7). Finally, by removing primer peaks (>50 bps) and spurious peaks (relative abundance > 0.5 %), the T-RFLP profile was figured out. The relative peak area of a single T-RF was calculated according to the following equation:

Ap = Ni/N×100                                                                                                             (2)

Where the Ni and N are the peak areas of the single and total T-RFs, respectively. T-RFs were analyzed by MiCA3 PAT and referred to the website (http://mica.ibest.uidaho.edu/pat.php), from which bacterial community structure spectrums were constructed.

2.8. CCA analysis:

Correlation analysis was conducted on species composition and variables factors by using canonical correspondence analysis (CCA) with CANOCO (Windows, 4.5 package). Parameters like COD, pH, decolorization rate, evenness and Shannon diversity index was selected as variable factors for the seven different samples that were taken from reactor. The CCA biplot characterize the biological communities relative to the selected variables. The arrows of the biplot represents the variable factors, the length of arrow represent the maximal value at its tip [1920].

  1. 117: not clear what a “bacteria liquid coated sample” is

Response: Thanks for the query. Here, the “bacteria liquid coated sample”, which is actually a spelling error, should be “bacterial solution”. Changes have been incorporated in revised manuscript for the better understanding of the readers.

  1. 120: “Different forms of single colonies were selected” This seems to imply that not all colonies were picked. Therefore, you’re getting a very biased subset of the community, not only because you’re missing all non-culturable taxa, but also picking a subset (by eyeballing different- (or similar-) looking colonies). Wording later in the manuscript makes it sound like these were the only strains (or at least the only culturable ones), but if only a subset was picked, this is in doubt

Response: Thanks for your query. The whole method has been explained above. I hope it has cleared over experimental design and writeup presentation.

  1. 122: “dashing operation” … does this mean streak plating?

Response: Thank you for the comment and highlighting the error. Yes, the term “dashing operation” means the streak plating method. Apologies for the inconvenience, subjected phrase has been removed from revised manuscript.

Table S2: This table makes it appear that your 9 isolates matched the GenBank strains exactly (it’s later made clear that this is not the case). You should label the GenBank strains as something like “Closest Relative” and give the % similarities in the table. Also, some of your descriptors (“Big” and “Thick”) should be better defined, e.g., with numerical ranges. Also, not clear what “Color difference” means and “Water solubility” seems to refer to color?

Response: Thank you for the comment. Sorry for the mistake on terms of “Color difference” and “Water solubility” which should be “Color difference between up and down side of the colony” and “Water soluble pigment”, respectively. In response to your advice, the unprofessional and unprecise morphological description was modified carefully and complete the data on the homology of strains. For specific changes, please refer to Table S2 in the supplementary material.

Table S3: “Urease”?

Response: Thank you for the comment. Urease functionally belong to the superfamily of amidohydrolases and phosphotriesterases. Ureases are found in numerous bacteria, fungi, algae, plants, and some invertebrates, as well as in soils, as a soil enzyme.

  1. 128 (Fig. 1 legend): legend description needed for right-hand photo

Response: Thank you for the comment, suggested recommendation has been incorporated in the revised manuscript. The right-hand figure is the SEM photo of one of the isolated strains.

  1. 131: Sounds like you extracted sludge DNA but didn’t use it…but later it’s clear that you did sequence from this extracted DNA. Clarify wording here.

Response: Thank you for the suggestions. Subjected phrase has been modified as per the reviewer’s guidelines in the revised manuscript.

Fig. S1: Describe the gel image in figure legend or omit. Tree is probably sufficient here

Response: Thank you for the comment. Gel image (Fig. S1) has been omitted in the revised manuscript.

  1. 147: It appears that each treatment (different dye in columns) was unreplicated. Clarify if this is incorrect. This makes some of your interpretations weak (see Results/Discussion comments)

Response: Thank you for the critical analysis. Here, no repetitions were performed in terms of column treatment or acclimatization. Actually, this test was performed just for the acclimatization purposes, and we tried to put bacterial strains under environmental stress to provide the metabolic pathways for their growth. Culturing in column was conducted to obtain the functionalized strains (acclimatization by azo dyes), this step specifically acclimatized the strains for decolorization test and to isolate the desired bacteria community.

  1. 154: What does the 1 cm thickness here refer to?

Response: Thank you for the comment. Here the 1 cm thickness refers to the cylindrical wall thickness of the column, modification has been incorporated in the revised manuscript.

  1. 154: I’m confused by this sentence. It sounds like your 9 isolates were added in addition to the bacteria in the sludge to the Winogradsky columns. But, if so, which isolates to which columns (dyes or dye combos)? Later in the manuscript it sounds like this was not the case, and instead the 9 strains were isolated from the bacterial assemblages in these columns after changes (“succession”) during the incubation of these columns. But I’m not 100% sure which it is.

Response: Thank you for the query. Yes, the later statement is correct, that all the strains were isolated form the bacterial colonies extracted after the column acclimatization.

Fig. 2: Need more description in legend. What are the 8 dye solutions (I count 7 from text)? What is the rainbow showing – light penetration?

Response: Thank you for the comment and queries. Yes, the rainbow shedding shows the light penetration, total 8 different dyes solution were applied at the same time namely; Control, Methyl Orange, Methyl Red, Alizarin yellow R sodium salt, Mixture, Tartrazine, Orange G and Orange I. To further clarify, names of the different dyes solution and figure legends have been modified in the revised manuscript.

  1. 173: “restriction” enzyme. I’m not clear why the T-RFLP work was done. Much more information was derived from the DNA sequencing (if I’m understanding the Methods description correctly). The T-RFLP is used to show assemblages in the colonies were “different” but that’s also clear from the CCA of sequencing data.

Response: Thank you for the comment and queries. As you said, T-RFLP technology is not as good as next-generation high-throughput sequencing technology in species resolution, but as a fluorescent-labeled primer-assisted enzyme digestion recognition technology, it is still stable and accurate in providing information of the the main microbial community of environmental samples. This is also the best way under our current conditions. We hope the reviewers can understand it.

  1. 176: This sounds like each reaction mixture contained multiple templates. I believe you mean microliters (ul) here, not micrometers (um). The volumes don’t add up to 25 ul.

Response: Thank you for the comment and highlighting the mistake. Yes, these volumetric measurements are in microliters (ul) units. Suggestion has been incorporated throughout the revised manuscript.

  1. 196-197: This sentence could instead go in to the Fig. 6 legend

 Response: Thanks for the comment, reviewer’s suggestion has been incorporated in the revised manuscript.

Results & Discussion

  1. 202: I’m confused by the “1: 8: 8 days” These apparently refer to sampling times at 1 day, 8 days, and ???

Response: Thank you for your time and the critical analysis on our research work. Here, the ratio of 1:8:8 represent the time interval of sampling to measure the acclimatization stage. For that purpose, sampling was divided into three groups on the basis of time, 1st group samples were taken right after 24 hrs (1 day), however, 2nd and 3rd group sampling was performed after 8 days. Here, the samples of 3rd group were take only to detect the acclimatization based absorbance removal. Additionally, Figure 3 represent the decolorization of these three groups, i.e. First Group (1 day), Second Group (8 day) Third Group (8 day). According to the reviewer’s suggestion changes have been incorporated in revised manuscript (Color Red), and are also given below:

 The first group of samples was taken after 24-hr, and the samples of the second and third groups were taken after eight days. The acclimatization of third group was performed to analyze the absorbance.

Fig. 3: What are the “three circles”? Does this refer to the 3 sampling times? Also, “Miture” to “Mixture”

Response: Thank you for the comment and highlighting the typing mistake. Yes, the term “three circles” refers to the 3 different sampling times. i.e. 1:8:8. Moreover, the highlighted spelling mistake has also been incorporated in the revised manuscript.

  1. 220: What is “smooth” decolorization?

Response: Here, the term “smooth” means to sat the uniform decolorization. For better understanding of the readers, modification has been incorporate in the revised manuscript.

  1. 235: Is this really a “rate” or rather a % or “efficiency”?

Response: Thank you for the comment. This actually the percentage (%) of decolorization. The subjected phrase has been modified in the revised manuscript.

  1. 237: Wording could be clearer: “resulted in lower biomass production” More importantly, what is the evidence for this?

Response: Thank you for the comment and analysis. Subjected phrase has been modified in the revised manuscript as per the reviewer’s suggestion. We have deleted the original speculation “The higher concentration of dye was toxic to microorganisms and resulting into the less production biomass.” which is our speculation without direct data from the present study.

Fig. 4: Are these proportions based on (number of colonies of strain X)/(totaI number of colonies from a Winogradsky column)? If so, make this clearer in Methods section. I believe the results here are based in single replicates (a single column), but correct this in Methods if this is incorrect.

Response: Thank you for the comment. Yes, these results are based on single replicates.

  1. 249, 252: some of your rank orders are not exactly correct

Response: Thank you for the suggestions. Here, bacterial strains have been explained based on their abundance and higher outcomes. So, it was not convenient to explain every strain and their outcome as it can be seen from the figure 4. In order to make it more feasible, possible changes have been done in the revised manuscript.

  1. 254-255: Unsure what is meant by “Comprehensive physiological and morphological analysis further showed that the isolates obtained in this study were mainly Bacillus sp., Stenotrophomonas sp., Paracoccus sp., and Pseudomonas sp.” I thought the colonies were identified by DNA sequencing. Would the physiological and morphological analyses change your identifications?

Response: Thank you for the comment and queries. Here, the physiological and morphological analyses do not represent the change in identification, but indicate their percentage of contribution against each colony.

  1. 256-261: Although both of these (phenotypic changes, evolutionary adaptation) may have occurred, this study only shows changes in bacterial assemblage structure during incubation.

Response: Thanks for the comment. Yes, in this study we mainly focused on structure based bacterial assemblage against different dyes decolorization. However, on the basis of previous literature, it can be seen that phenotypic changes and evolutionary adaptation may also occurred under the domestic conditions

  1. 265: This should be in Table S2 and made clear that these GenBank taxa are just the closest sequence matches, rather than the very same strains

Response: Thanks for the comment. We made clear notion in Table S2 that these GenBank taxa are just the closest sequence matches rather than the very same strains.

  1. 268: should be “Bacillus anthracisstrain ####”

Response: Suggested recommendation has been incorporated in the revised manuscript.

  1. 282: some of these likely would not be new species, esp. the ones with very high (99%) sequence similarity

Response: Thank for the comment. These line shows the major contaminants in terms of various bacterial genera, against the textile wastewater treatment that were isolated with phylogenetic similarity of more than 95 % in each case.

  1. 288-289: these are not phyla

Response: Thank for careful check. The “phyla” was changed to “genera” in the manuscript.

  1. 290: not the most dominant in all 4 of these treatments

Response: Thank for careful check. Azo dye of “tartrazine” was removed, and Bacillus cereus strain 1FFF dominates in other 3 treatments. Suggested recommendation has been incorporated in the revised manuscript.

  1. 292-297: Where are these %s coming from? Don’t seem to agree with those in Fig. 4.

Response: These %s is the raw data based on which the Fig. 4 was established. Emphasizing the quantity here is to clear bacterial community scale in different treatment.

  1. 297: Meaning of this sentence is unclear: “In comparison with bacterial community and azo dye structure, similar phyla composition was observed both in columns and in the structures of Methyl Orange and Methyl Red.”

Response: We corrected the writing as “In comparison with bacterial community and azo dye structure, similar microbial phyla structure was observed in columns feed with Methyl Orange and Methyl Red which have similar chemical structure to each other.”

  1. 313: text says 13 phyla, figure legend implies 16

Response: Thanks for the kind reminding. From the perspective of bacterial phyla, the original dewatered sludge contains 13 different types of bacteria on phylum level, the legend of Figure 5 only implied the phylum of all the strains observed in all treatments, and from the raw data set of the pie figure of the control, 13 phyla were found on the microbial level.

  1. 335: Is “Actinobacteria” here a typo? Are you trying to say that these two dye treatments were different, especially in that they had higher %s of Actinobacteria?

Response: We are very sorry that our negligence caused you trouble. This is indeed our writing error, “Actinobacteria “has been deleted here.

  1. 343: most of this paragraph is redundant with the prior paragraph

Response: Thank you for the constructive suggestion. We simplified this part, deleted this paragraph and reorganized the rest into the last paragraph to make the description here more fluent like “However, the decolorization of Tartrazine and Orange G with relatively complex structures was specifically attributed to the Actinobacteria and Proteobacteria phyla, respectively, while but the Actinomycetes was accounted as a dominant bacterium for Hydrazine decolorization. Because of different azo dye structures, the corresponding phyla structures and their abundance formation during the acclimatization process were different from each other. Furthermore, the abundance of phyla in the mixed dyes column decreased up to a certain extent as compared to the single dye columns. It is probably attributed to the toxicity of mixed azo dyes that was higher than any of the single dye, resulting into the reduced microbial activity.”

  1. 349: “Hydrazine” comes out of nowhere. Citation needed?

Response: Thanks for mentioning this. We have cited relevant report to support our point of view here.

  1. 350: this statement (and many others like it) cannot be supported statistically because no replicates. It’s possible that stochastic factors, rather than the dye in each column, were the cause of differences in diversity, community structure, etc. At a minimum, this limitation of the study must be mentioned in the Discussion.

 Response: Thank you for the constructive suggestion, recommendation has been incorporated in the discussion. The lack of replicate settings does weaken the original conclusion that the bacterial community structure in the sludge depends on the chemical structure of the dye molecules used for domestication. Even if there are repeated settings, the original conclusion is still too confirmative in the presence of existing data, therefor we have weakened the affirmative statement, only pointed out this phenomenon and possible reasons. Please check the specific modification in the manuscript.

  1. 380-386: redundant with Methods (or should be moved there)

Thanks for your comment to modify this part. The required part has been moved to methodology section and changes have been highlighted in manuscript.

  1. 384: Densities not actually measured. Perhaps “relative abundance” would be OK here

Thanks for your suggestive feasible term to use here. Its right, the relative abundance is actual term to use her. Changes have been done in manuscript.

Fig. 6: What are D-1 through D-7? Define each one in legend. Change “principle” to “principal”.

 Response: D1-D7 indicated the microbial community structure information in the sludge domesticated by different dyes, which is actually the data set in figure 4 being directly used as the species distribution information in the CCA analysis, combined with the environmental factor information to perform the CCA biplot analysis to reflect the similarity of microbial habitats and main influencing factors represented by different domestication treatments. The requested changes have been done in revised manuscript.

It’s odd that your variables here are a mix microcosm conditions, bacterial assemblage characteristics, and effects on dyes, i.e., a mix of causes and effects

Response: Thank you for the comment.

CCA is a sorting method developed based on correspondence analysis. It combines correspondence analysis and multiple regression analysis, and each step of calculation is regression with environmental factors, also known as multiple direct gradient analyses. This analysis is mainly used to reflect the relationship between the flora and environmental factors whch is based on a unimodal model. Analysis can detect the relationship between environmental factors, samples, and flora, or the relationship between any two of them.

Here we hope to comprehensively define each type of domestication treatment based on the environmental factors (such as pH variation) and ecological functions (such as decolorization performance and COD removal rate), combining with the information on the microbial community structure, to clarify the microbial habitat characteristics and similarity distributions represented by the sludge domesticated with different azo dyes, as well as the key influencing factors. Here, biological characteristic parameters (evenness index and Shannon index) and ecological functions (in a sense, as the reviewer said, ecological functions are determined by variables such as environmental biological factors and non-biological factors) were also used as environmental biological factors, which was in the hope that the dimensions of the indicators that define the characteristics of the habitat are more extensive (the relevant environmental factors are somewhat less). And after data pre-assessment, the relationship between various types of data presents a unimodal model, which is suitable for CCA analysis, so this kind of analysis is performed here. Furthermore, according Figure 6, it can be seen that the contribution of different variables to defining the characteristics of the habitat (as can be seen by the length of the arrow line) and the positive and negative correlations are different, indicating that in this research situation, it can be used to comprehensively reflect the characteristics of different habitats.

Of course, in future research, we will focus on sorting out the logical relationships of different data sets and conducting more reasonable analysis. We hope our explanation will help your understanding, and thank you again for your constructive comments.

  1. 388: “all variables have same influence on microbial communities” See comment above about cause and effect

    Response: Thank you for the reminding. This part of the statement is indeed inaccurate, and it is not what we want to explain. It has been revised in the manuscript to “Under the influence of different environmental factors, different acclimated sludges reflected different habitat characteristics and similarity distributions, among which the degree of influence of various environmental factors was also different.”

  1. 390: “differ not only from all other variables” I think the intent here is to say “differ not only from all other dye treatments”

Response: Sorry for the wrong term to present our point. You’re right. The intent here was to say differ not only from all other dye treatments.

  1. 391: what is meant here by “in-depth analysis”?

Response: Thank you for the comment. Here, the term “in-depth analysis” means to say the further comparison of these habitats based on the CCA analysis.

  1. 394: “centrifugal”?

Response: Sorry for the Irregular expression, should be “arrow line” here.

  1. 406: by “density” do you mean the most species?

Response: Yes, the term “density” here means to say the more or higher number of species.

  1. 409: these are not isolates

Response: Thanks for the careful checking. The term “Isolates” is indeed incorrect here, which should be “bacterial community” according to the data we adopted in CCA analysis.

  1. 414: “followed the same mechanism” Identifying the mechanisms for decolorization was a stated goal of the study. Although you may have identified some of the microbial players in the process, I don’t think you can say much about the actual mechanisms

Response: Thank you for the comment. The argument here does lack data support, and as an inference, there is no corresponding research support. We have deleted this part of the content.

  1. 420: “composition of the community structure” Composition is part of the structure, so this should be re-worded

Response: Thank you for the comment. Highlighted phrase has been modified in the revised manuscript as per the reviewer’s suggestion.  

  1. 422: As far as I can tell, the data in Fig. 7 come from a single sampling time so really can’t say anything about succession. However, this may be incorrect because some sampling was done at three time points. In any case, it should be made clear in the Methods section from which time point(s) the data for Figs. 4-7 are derived.

Response: We are sorry that we did not specify the sample source of the data in the methods section. Here we use domesticated sludge of the last cycle (the sample on the 21st day after the acclimation) as the sample for microbial community structure analysis. We have add this information in the method section.

The succession of the microbial community we want to reflect here is relative to the microbial community in the original dewatered sludge without the azo dye acclimation, that is to say, the microbial community in the original dewatered sludge is used as a starting point to investigate whether there is a differentiated succession of the microbial community, and whether this succession path or characteristic has an apparent relationship with the similarity and difference of the molecular structure of different dyes in the sludge after acclimation of different azo dyes.

  1. 424: no evidence for this statement

Response: Thank you for the comment. Your opinion is very correct. Here we have made an overly positive judgment on the relationship shown by the data. It is indeed a lack of evidence of biological relationship. It can only be a possibility. We have also adjusted the expression in the summary and discussion as a kind of May be hypothesized. We have deleted the discussion here.

Fig. 7: CK is now D8?

Response: Thanks for your keen thought on this figure. D1-D7= the bacterial community structure of sludge acclimated by Methyl orange, methyl red, orange I, orange G, tartrazine, alizarin yellow R sodium salt and mixture, and CK=D8.

  1. 429: No appendix?

 Response: Sorry for missing file. The related information has been updated into supplementary information as Table S3.

Conclusions

  1. 431: “investigated the microbial dynamics of the azo dyes having different chemical structures” Wording here could be clearer

Response: Thank you for the comment. Highlighted phrase has been modified in the revised manuscript as per the reviewer’s suggestion.

  1. 440: “contribution rate of Shannon diversity index” Awkward phrasing

Response: Thanks for the comment and suggestion. Changes has been incorporated in the revised manuscript for the better understanding of the readers.

  1. 422: meaning unclear

Response: Thank you again for highlighting the deficiencies in our manuscript. The subjected lines have been modified in the revised manuscript.

  1. 443-445: this speculation should be removed (see comment above)

Response: Thank you for comment. Apologies for the inconvenience, instead of removing the lines, we restructured the phrases for the better understanding of the readers and the modifications have been highlighting in the revised manuscript.

Round 2

Reviewer 1 Report

The manuscript has been deeply improved although it keeps some issues to revise.

Since the study claims the correlation between the azo dye structure and the class of bacteria to explain the conversion achieved, it is mandatory to show the formula and chemical structure of the azo dyes inspected.

Line 66. "its believed" must be "it is believed".

Line 169. "10min" still needs a space... "30mL" in line 178, too, as well as "100mg/L to 400mg/L" in line 266, 100mg/L in line 270.

Line 186. "by using the following equation:" must be "by using equation 1:". Use the same format in line 209.

Why some times do you write "24-h"?

Line 243. "the combine..." must be "the combination..."

Author Response

Response to Reviewer 1

The following is a detailed response to the reviewers’ comments and suggestions.

Note: “Detailed Responses to Reviewers’ Comments” were marked with BLUE and all modifications and changes made in the revised files have been HIGHLIGHTED.

The manuscript has been deeply improved although it keeps some issues to revise.

Thanks a lot for your devoted time and efforts on this paper in order to make it presentable. We already have done many changes according to the suggestions of worthy reviewers, hope this time you will also like it.

Since the study claims the correlation between the azo dye structure and the class of bacteria to explain the conversion achieved, it is mandatory to show the formula and chemical structure of the azo dyes inspected.

Its good point to highlight here. It was our mistake to not add the table presenting the chemical characteristics of investigated azo dyes. Table S1 has been added in supplementary materials to present your idea. Moreover, addition of this table has been likewise highlighted in manuscript.

Line 66. "its believed" must be "it is believed".

Thanks for your remarks. The changes have been done in the manuscript according to your recommendation.

Line 169. "10min" still needs a space... "30mL" in line 178, too, as well as "100mg/L to 400mg/L" in line 266, 100mg/L in line 270.

Thanks for mentioning typo error in the manuscript. All the changes have been done according to your presented point. We hope there’re not more.

Line 186. "by using the following equation:" must be "by using equation 1:". Use the same format in line 209.

Thanks for your comment at this stage. It was a major mistake which was not highlighted. Now, changes have been done in the manuscript.

Why some times do you write "24-h"?

Thanks for your mentioning. All typo errors have been arranged. Plz check the manuscript with highlighted parts.

Line 243. "the combine..." must be "the combination..."

All authors highly appreciate your comments. The modified version is highlighted with your suggested comments. Hope you will like it this time.

Reviewer 2 Report

Several improvements to the manuscript were made in the revision, although many mistakes and inadequacies remain. Some the remaining issues are simply because the authors made clarifications to the reviewers, but did not clarify in the actual manuscript. Comments on each section below:

Introduction

l. 63: reference irrelevant

l. 85: "inactivated bacteria preferred"...why state this when your study didn't do this (or why didn't you deactivate?)

l. 107: "in" to "isolated from"

l. 108: odd wording

Methods

l. 148, 168, 169, 178, 190, 200, 266, 270 and Fig. S2: other reviewer suggested separating number from units, which the authors said they'd correct...failed to do so in all the spots above

l. 150: similar to above, still have units such as hr and sec

Fig. 2: SEM still not described or referred to anywhere

Fig. S1: gel still here (vs. response to reviewer)

l. 185: not a rate (see also Fig. S2; see previous comments by both reviewers)

l. 194: TRFLP not needed; HTS data are more informative

Results

l. 232: "1:8:8" is still confusing and should be omitted (just explain better in text what when sampling was done); the  "3 groups" idea is also confusing here

Fig. 3: Legend is still confusing...what the "circles" refer to is still confusing to reader

l. 240: as compared to what...methyl orange?

l. 242: should be in Methods. It sounds like there was two samplings at 8 days, but used for different measurements...but both show up in data (e.g., Fig. 3). Still confusing.

l. 278: mistake here. B. cereus 1FFF looks to be ~2%, not 30.4% in methyl red

l. 279: "followed by..." Need to clarify. In which dye mesocosm?

l. 282: "followed by..." Incorrect. B. cereus clearly not 3rd most abundant (it's #1)

l. 295: Why don't these strains line up with your Fig. 4? Disagrees with Table S1 and Fig. 4.

l. 296: What happened to T2?

l. 298: disagrees with Table S1

l. 299: disagrees with Fig. 4

l. 301: disagrees with Table S1

l. 302: What isn't S. malto twice in FIg. 4? Text and Table S1 appear to disagree for T-7

l. 308: missing some (e.g., Klebsiella)? Or does "contaminants" here mean something other than "isolates"?

l. 311: > 95% does not mean a new species. Maybe you could use along with "<99%"

l. 319: these are not phyla (mean "taxa"?)

l. 322: mistake here...3 dyes or mixtures listed, followed by 4 %s

Fig. 5: Presumably this is your DNA sequence data? If so, what does the "uncultured" pie piece actually mean? "Unidentified" (no database match)?

l. 358: Why aren't the diversity index values provided somewhere (even in supplemental materials)? Also, it does not appear that you have before-after data for any samples (i.e., dye-added treatment or the "raw sludge") so can't really talk about diversity declining through time, succession, etc. Description of when the samples were taken for data in Figs. 5-7 needs to be clarified. Is raw sludge from 8 days, or near beginning (Day 1)? Ideally you'd have both t=0 and t=final for both dye-treatments and "control"

l. 406: CK = D8? Why use two labels?

l. 408: Still not clear to reader what the "in-depth analysis" here refers to.

l. 438-442: "succession" not backed up by data or analyses

Conclusions:

l. 451 & 457: "compared with original sludge" See comments above about timing of samples and reporting of diversity metrics

Author Response

Response to Reviewer 2

The following is a detailed response to the reviewers’ comments and suggestions.

Note: “Detailed Responses to Reviewers’ Comments” were marked with BLUE and all modifications and changes made in the revised files have been HIGHLIGHTED.

Introduction

  1. 63: reference irrelevant

Thanks for your point. In this study, author established an anaerobic sequencing batch reactor (ASBR) fed with increasing methyl orange (MO) concentrations (from 25 mg l−1 to 500 mg l−1) in this study. The relationship between MO decolorization capacity and the microbial community structure was evaluated. More than 85% of COD and 75% of MO were removed during the whole operation period. 

  1. 85: "inactivated bacteria preferred"...why state this when your study didn't do this (or why didn't you deactivate?)

Thanks a lot for your devoted time and efforts on this paper in order to make it presentable. We are sorry that the description here troubles you. According to our literature research, in addition to the direct degradation of azo dyes, microorganisms can also remove azo dyes by adsorption and precipitation, mainly due to the existence of various related functional groups on the cell surface. However, this part of the expansion is indeed too cumbersome, which can easily lead to a lack of focus on the problem orientation, making readers think that we are focusing on microbial adsorbents. According to your suggestion, we have deleted the part of the “Moreover, inactivated microorganisms as adsorbents of decolorized azo dyes are preferred due to: (1) no nutrient requirement; (2) long-term preservation; (3) regeneration through organic solvents and surfactants.” in the introduction.

  1. 107: "in" to "isolated from"

Thanks a lot for your correction. Its right. Have done the changes in the manuscript.

  1. 108: odd wording

We really appreciate your suggestions. Some changes have been done to highlight the real meaning of the words. We hope you will like it.

Methods

  1. 148, 168, 169, 178, 190, 200, 266, 270 and Fig. S2: other reviewer suggested separating number from units, which the authors said they'd correct...failed to do so in all the spots above

Thanks for the corrections. There were typo errors. Have been changed. We hope you will not face this problem now.

  1. 150: similar to above, still have units such as hr and sec

Thanks for your mentioned errors. All changes have been added in final manuscript.

Fig. 2: SEM still not described or referred to anywhere

Thanks for highlighting this part. This part refers to strain isolation experimental part, which has been referred in the final manuscript.

Fig. S1: gel still here (vs. response to reviewer)

Thanks for highlighting the issue again. That’s the error, which was not corrected last time, apology for that. Changes have been added in the final version.

  1. 185: not a rate (see also Fig. S2; see previous comments by both reviewers)

Thanks for the statement. The changes have been added in the manuscript.

  1. 194: TRFLP not needed; HTS data are more informative

Thank you for your question. As we have replied before, HTS is indeed the most mainstream microbial molecular ecology analysis method, which can provide comprehensive information on microbial diversity and functional genes. T-RFLP has also been successfully applied to the analysis and comparison of various microbial communities, to study the diversity and structural characteristics of microbial communities, and its advantage lies in its ability to quickly distinguish the distribution and combination characteristics of dominant bacterial communities in different samples. The species resolution of TRFLP is definitely not as high as that of HTS, but combined with the existing conditions of our laboratory and the purpose of this research, T-RFLP technology is used.

【1】Lee, S. H., Lee, H. J., Kim, S. J., Lee, H. M., Kang, H., & Kim, Y. P. (2010). Identification of airborne bacterial and fungal community structures in an urban area by T-RFLP analysis and quantitative real-time PCR. Science of the total environment, 408(6), 1349-1357.

【2】Karczewski, K., Riss, H. W., & Meyer, E. I. (2017). Comparison of DNA-fingerprinting (T-RFLP) and high-throughput sequencing (HTS) to assess the diversity and composition of microbial communities in groundwater ecosystems. Limnologica, 67, 45-53.

【2】Kari, A., Nagymáté, Z., Romsics, C., Vajna, B., Kutasi, J., Puspán, I., ... & Márialigeti, K. (2019). Monitoring of soil microbial inoculants and their impact on maize (Zea mays L.) rhizosphere using T-RFLP molecular fingerprint method. Applied Soil Ecology, 138, 233-244.

Results

  1. 232: "1:8:8" is still confusing and should be omitted (just explain better in text what when sampling was done); the "3 groups" idea is also confusing here

Thanks for your comment. The sampling time according to time period has been explained in text. The third group (absorbance study) has been placed in next line to separate from first two group. We think its clear now.

Fig. 3: Legend is still confusing...what the "circles" refer to is still confusing to reader

Thanks. Figure 3 is presenting the percentage decolorization of different azo dyes for three selected groups. Acclimation period of every group has been highlighted in the legend. The word circle has been replaced with group to clear the meaning.

  1. 240: as compared to what...methyl orange?

Thanks for your comment. The performance was presented here in comparison with all other targeted dyes, which has been written in the final manuscript.

  1. 242: should be in Methods. It sounds like there was two samplings at 8 days, but used for different measurements...but both show up in data (e.g., Fig. 3). Still confusing.

Following your suggestion, we deleted this part of the content “Mixture of all dyes were also utilized to analyze the combination effects on decolorization rate percentage” which may cause confusion in the results. The acclimation period and the decolorization determination are synchronized. There are 7 chromium settings, including 6 single dyes and a mixed dye domestication setting. The decolorization determination is performed during the acclimation, a total of three cycles (approximately 7 days in a cycle).  The decolorization rate was measured at the end of each cycle as shown in Fig. 3. This part has also been described in the method part as “The dewatered sludge (100 g) was inoculated in acclimatized medium (800 mL) containing 30 mg/L of single azo dye each in six differ columns and 1 column with mixture of 6 azo dyes mentioned above under room temperature without specific illumination. After complete decolorization, the supernatant was removed, and new culture medium and dye was added for the next acclimatization cycle, totally 3 cycles were conducted. At predetermined time intervals (about 7 days a cycle), 2 mL samples were taken, filtered, centrifuged and the dye concentration was spectrophotometrically determined (UV2300 spectrophotometer China).”

  1. 278: mistake here. B. cereus 1FFF looks to be ~2%, not 30.4% in methyl red

Thanks for mentioning the big mistake. The point was placed wrong in the digits. It was 3.04% instead of 30.4%. The statement has been changed and rewritten in the final manuscript.

  1. 279: "followed by..." Need to clarify. In which dye mesocosm?

Thank you for your correction. We rewrite this part as “Bacillus cereus strain 1FFF (Firmicutes) have proportions of 60%, 38.63% and 30.4%, respectively, followed by Exiguobacterium aestuarii strain YS-6 (Firmicutes) and Klebsiella variicola strain RVEV3 (Proteobacteria) in orange I, alizarin yellow R sodium salt and mixture, respectively).” Here refers to the microcosms domesticated by dyes orange I, alizarin yellow R sodium salt and dye mixture.

  1. 282: "followed by..." Incorrect. B. cereus clearly not 3rd most abundant (it's #1)

Sorry for the oversight here, the correct statement should be as “Bacillus cereus strain 1FFF (Firmicutes) have proportions of 60 %, 38 % and 35 %, respectively, followed by Exiguobacterium aestuarii strain YS-6 (Firmicutes) and Klebsiella variicola strain RVEV3 (Proteobacteria) in orange I, alizarin yellow R sodium salt and mixture, respectively). 

  1. 295: Why don't these strains line up with your Fig. 4? Disagrees with Table S1 and Fig. 4.

The microbial strain information in Table S1 and Fig. 4 is based on cultivable strains, that is, quantitative statistics and strain identification are performed based on strains with similar morphological, physical and chemical properties isolated from different acclimation amples, and strains with the closest genetic relationship were obtained through phylogenetic analysis. The strain information in Fig. 5 was based on TRFLP analysis. The comparison database is different, and there were more operating taxa units from the data of TRFLP. Therefore, the number of strains reflected by TRFLP was more than that based on cultivable strains comparison results are also inconsistent. The comparison database for TRFLP data analysis of environmental sample and the identification of isolated strains is different, so the strain information obtained by these two methods were not completely consistent, especially at the genus level, but there was still a certain consistency at the phylum level.

  1. 296: What happened to T2?

Sorry for the negligence here. The description of isolate T-2 has been added in the manuscript as “Strain T-2 was highly homologous to Klebsiella variicola strain RVEV3, with a sequence similarity of 95%.”

  1. 298: disagrees with Table S1

Thank you for pointing out this tiny error carefully. I think what you are referring to here is that the description of isolate T-5 is inconsistent with Table S2 (not Table S1) and the correction has been made in the manuscript as “The phylogenetic similarity between T-5 and Bacilus anthracis strain WY2 was only 83%.”

  1. 299: disagrees with Fig. 4

Correction has been made in the manuscript as “The strain T-6 was highly homologous to Pseudomonas geniculate ka38 with a phylogenetic similarity of 96%.”

  1. 301: disagrees with Table S1

Correction has been made in the manuscript as “Additionally, about 97% and 99% of the phylogenic similarity was observed between T-7 and Bacillus cereus strain BVC79, and T-8 and Stenotrophomonas maltophilia T-11, respectively.”

  1. 302: What isn't S. malto twice in FIg. 4? Text and Table S1 appear to disagree for T-7

We are very sorry for the mistake, this is a duplicate, the information in Table S1 is correct, and the text description is wrong. Correction has been made in the manuscript as “Additionally, about 97% and 99% of the phylogenic similarity was observed between T-7 and Bacillus cereus strain BVC79, and T-8 and Stenotrophomonas maltophilia T-11, respectively.” This is the same error appeared in line 301.

  1. 308: missing some (e.g., Klebsiella)? Or does "contaminants" here mean something other than "isolates"?

Here "contaminants" is an inappropriate expression. What we want to express is "isolates", which refers to the main azo dye decolorizing strains isolated in this study. However, combined with the context, the expression here seems redundant, so we deleted this part from the manuscript.

  1. 311: > 95% does not mean a new species. Maybe you could use along with "<99%"

Thank you for your professional advice and accurate understanding of our willingness. here is insufficient evidence to confirm that the isolates are new species. We adjusted the expression as “Total 7 isolates out of 9 obtained in this study showed a similarity percentage of less than <99% with the standard strains. Consequently, these strains can hardly be claimed as new species having applicability in taxonomic study.”

  1. 319: these are not phyla (mean "taxa"?)

Correction has been made in the manuscript as “The dominant population were Bacillus cereus strain 1FFF, Klebsiella variicola strain RVEV3 and Exiguobacterium aestuarii strain YS-6.”

  1. 322: mistake here...3 dyes or mixtures listed, followed by 4 %s

Correction has been made in the manuscript as “The most dominant group in Orange I, Alizarin yellow R sodium salt and mixed dyes was Bacillus cereus strain 1FFF with the compositional proportion of about 60 %, 38 % and 35 %, respectively.”

Fig. 5: Presumably this is your DNA sequence data? If so, what does the "uncultured" pie piece actually mean? "Unidentified" (no database match)?

As you said, “Unidentified” should be the most accurate notion to the term “uncultured” here. As shown in Fig. 5, there are many sequences in our TRFLP data that cannot be identified to any possible microbial population, that is, the species information cannot be determined through sequence alignment, therefore cannot be identified according to the existing database, which normally refers to potential and uncultured microbial populations in the environment.

  1. 358: Why aren't the diversity index values provided somewhere (even in supplemental materials)? Also, it does not appear that you have before-after data for any samples (i.e., dye-added treatment or the "raw sludge") so can't really talk about diversity declining through time, succession, etc. Description of when the samples were taken for data in Figs. 5-7 needs to be clarified. Is raw sludge from 8 days, or near beginning (Day 1)? Ideally you'd have both t=0 and t=final for both dye-treatments and "control"

Thank you for your careful review and suggestions. We have added the sampling time for microbial molecular ecology analysis in the method section as “Partial sludge samples of raw dewatered sludge and the ones after acclimation (21 days) were sampled and kept at -20℃ for T-RFLP analysis.” As described in the introduction, one of the purposes of this study is to investigate the succession path of the microbial community in dewatered sludge exposed to different azo dyes with comparable structural differences, that is, what is the characteristics of the changes in the microbial community structure during the acclimation, like is there any azo dye structure-dependent success path of the microbial community. Therefore, from the perspective of our research objectives and experimental settings, the control was the raw dewatered sludge before acclimation, while the treated samples were the sludges acclimated for 21 days with different single azo dyes and mixed azo dyes. In this study, raw dewatered sludge is the dye-treatment of “t=0”, and sludge samples after 21 days acclimation is xx is the dye-treatment of “t=final”. According to the request, the diversity index of the bacterial communities of different treatments was added in Table S5 in the supplementary material.

  1. 406: CK = D8? Why use two labels?

Thanks. In this whole manuscript. D8 is presenting the CK. The changed have been done and named as CK now.

  1. 408: Still not clear to reader what the "in-depth analysis" here refers to.

The term keen or depth was used here. Sorry for trouble. The statement has been changed and written as “However, further analysis revealed that the D2 and D6, D1 and D7 were also from the similar territory, but D4 and D5 have independent habitat”.

  1. 438-442: "succession" not backed up by data or analyses

Thanks for highlighting this part. We think the word succession is leading to wrong concept. We think with deleting it, its presenting the fine part.

Conclusions:

  1. 451 & 457: "compared with original sludge" See comments above about timing of samples and reporting of diversity metrics

Thanks for highlighting this part. Here, author is presenting the idea of utilizing the dewatered sludge as compared with activated sludge.

Round 3

Reviewer 2 Report

Most of my objections have been adequately addressed. However, a few issues related to Methods and Results remain:

  1. Table S2 is not referred to in the text
  2. Diversity values (and new Table S3) are not mentioned within text of the manuscript
  3. Reference to Table S2 in sec. 2.2 is incorrect (should be Table S1)
  4. Table 3 is referenced in section 2.8, but it does not exist. Also, there are no Tables 1 or 2.

I believe the authors have addressed my questions about TRFLP use vs. HTS. I was confused, thinking that they had used both TRFLP (just for community fingerprinting) and HTS. I now realize that they just used TRFLP and obtained phylogenetic assignments as described in last sentence of sec. 2.7.

Author Response

Response to Reviewer 2

The following is a detailed response to the reviewers’ comments and suggestions.

Note: “Detailed Responses to Reviewers’ Comments” were marked with BLUE and all modifications and changes made in the revised files have been HIGHLIGHTED.

  1. Table S2 is not referred to in the text

Thanks for your comments. The table S2 was in the end of section 2.4. Moreover, the table S3 has been added in the same section in order to highlight the morphological and the physiological-biochemical characteristics of incorporated strains.

  1. Diversity values (and new Table S3) are not mentioned within text of the manuscript

Thanks to point out this issue. The table S3 was added in the manuscript but with typo error. I hope its clear now and your can easily find it.

  1. Reference to Table S2 in sec. 2.2 is incorrect (should be Table S1)

Thanks, as mentioned above, table S2 and S3 have been placed in section 2.4. where the incorporated strains are presenting first time.

  1. Table 3 is referenced in section 2.8, but it does not exist. Also, there are no Tables 1 or 2.

We really appreciate your considerate analysis on this paper. As mentioned above, it was a typo error. Now it has been removed. There’s no Table 1, Table 2 or Table 3.
